# Ultra-low and ultra-broad-band nonlinear acoustic metamaterials

Xin Fang[1], Jihong Wen[1], Bernard Bonello[2], Jianfei Yin[1] & Dianlong Yu[1]

Linear acoustic metamaterials (LAMs) are widely used to manipulate sound; however, it is challenging to obtain bandgaps with a generalized width (ratio of the bandgap width to its start frequency) >1 through linear mechanisms. Here we adopt both theoretical and experimental approaches to describe the nonlinear chaotic mechanism in both one-dimensional (1D) and two-dimensional (2D) nonlinear acoustic metamaterials. This mechanism enables NAMs to reduce wave transmissions by as much as 20–40 dB in an ultra-low and ultra-broad band that consists of bandgaps and chaotic bands. With subwavelength cells, the generalized width reaches 21 in a 1D NAMs and it goes up to 39 in a 2D NAM, which overcomes the bandwidth limit for wave suppression in current LAMs. This work enables further progress in elucidating the dynamics of NAMs and opens new avenues in double-ultra acoustic manipulation.

[1] Laboratory of Science and Technology on Integrated Logistics Support, National University of Defense Technology, Changsha, Hunan 410073, China. [2] Institut des NanoSciences de Paris (INSP-UMR CNRS 7588), Université Pierre et Marie Curie, (Box 840) 4, Place Jussieu, 75252 Paris Cedex 05, France. Correspondence and requests for materials should be addressed to J.W. (email: wenjihong@vip.sina.com) or to B.B. (email: bernard.bonello@insp.jussieu.fr)

Acoustic metamaterials[1–5] (AMs) are promising for many applications, including acoustic and vibration insulation[6–8], sound absorption[9], cloaking[10–12], sensors[4] and topological insulators[13]. Relatively broad low-frequency bands are desirable and most studies[1–18] in this field have focused on linear AMs (LAMs) based on the sub-wavelength locally resonant (LR) mechanism[1, 18]. However, LR bandgaps are generally narrow[2, 3]. The generalized width of a band is $\gamma = (f_{cut}-f_{st})/f_{st}$, where $f_{st}$ ($f_{cut}$) denotes the start (cutoff) frequency of the band. In theory[5] $\gamma = \sqrt{1 + m_r/m_b} - 1$ for the LR band-gap, where $m_r$ ($m_b$) is the mass of the resonator (the base media) in a meta-cell, for example, $\gamma \approx 0.22$ for $m_r = m_b/2$. Recent works couple the LR and Bragg bandgaps[19, 20] to obtain a width $\gamma = 0.71$ in a one-dimensional (1D) LAM[21] with a lattice constant $a \approx 2\lambda/5$ and $\gamma = 0.85$ in a two-dimensional (2D) LAM[22] with $a \approx \lambda/4$, where $\lambda$ refers to the wavelength at $f_{st}$. Therefore, obtaining a generalized width $\gamma > 1$ in LAMs remains a challenge. May nonlinearity help overcome this difficulty?

Similar to nonlinear electromagnetic metamaterials[23–26] where desired nonlinear responses have been demonstrated[27–31], nonlinear acoustic metamaterials (NAMs) deserve special attention. When acoustic waves propagate within a nonlinear acoustic medium, such as Fermi–Pasta–Ulam chains[32–34] or granular crystals[35–38], nonlinear phenomena including discrete breathers[39], solitons[40, 41] and bifurcations[42] can be observed. Acoustic diodes[36, 43, 44], rectification[45] and lenses[46] based on nonlinear media have been designed. However, the involved mechanisms hardly allow for simultaneous low-frequency and broadband properties; therefore, the discovery of new mechanisms is required for further progress.

For finite LAMs, bandgaps are stop bands; however, the broad passbands actually consist of dense resonances that localize energy to enhance incident waves. Recently, a mechanism was theoretically predicted in discrete NAMs[47–49]: the chaotic bands. Chaos is an aperiodic long-term behavior in a deterministic/ nonlinear system exhibiting a strong dependence on the initial conditions[50]. In NAMs[49], chaotic bands are those passbands in which an incident low-frequency periodic wave becomes a chaotic emerging wave, reducing wave transmission. The chaotic wave features a high-frequency continuous spectrum evidencing the dispersion of energy[47, 49]. These waves have lower amplitudes than the corresponding linear resonances; thus, NAMs can suppress wave propagation in the passbands. The wave suppression effect of the strong chaos is broadband and it depends on the frequency but not the width of the nonlinear LR bandgap[49]. Therefore, we can design a NAM with nonlinear meta-cells that generate ultra-low frequency but narrow linearized LR bandgaps. When strong nonlinearities occur, the passbands higher than these nonlinear LR bandgaps become chaotic and wave propagation is suppressed; an ultra-low and ultra-broad (double-ultra) band NAM is thus obtained.

In this work, we report on NAMs based on the chaotic band that achieves double-ultra band wave suppression. We design both a 1D NAM beam and a 2D NAM plate with periodic strongly nonlinear sub-wavelength meta-cells. When the strong nonlinearity appears, our experiments demonstrate that the NAMs substantially suppress wave propagation in the double-ultra bands. By combining frequency responses, bifurcations, Lyapunov exponents and different experiments, we describe the propagation of waves and demonstrate that the double-ultra effect is induced by the chaotic waves.

## Results

**NAM design.** As elucidated by the sketched band structure of the diatomic model (Fig. 1a, b), we propose 1D and 2D NAMs with

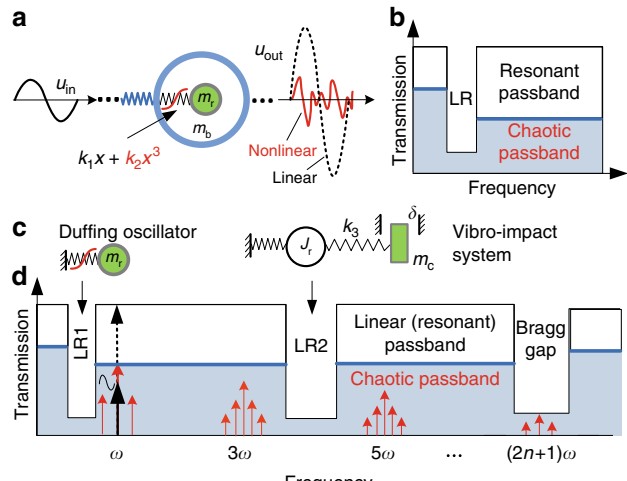

**Fig. 1** Schematic and conceptual diagram. **a** Diatomic NAM model composed of periodic linear base $m_b$ coupled with Duffing oscillators $m_r$ through the nonlinear spring $k_1x + k_2x^3$, where $k_1$ and $k_2$ are the linear and nonlinear stiffness coefficients, respectively. **b** Its band structure. Here, the passbands become chaotic bands, where a periodic input wave $u_{in}$ generates chaotic output wave ($u_{out}$ red) with an amplitude is much lower than that of the corresponding linear resonance ($u_{out}$ dashed black). **c** Two nonlinear sources in our NAM cell: Duffing oscillator and coupled vibro-impact system ($m_c$ couples to $J_r$ through a linear spring $k_3$ and with a clearance $\delta$). **d** Conceptual diagram of the double-ultra mechanism using chaotic bands. LR1 (LR2) represents the first (second) LR bandgap induced by the linearized Duffing and vibro-impact systems, respectively. The black (dashed black) arrow represents the input wave $u_{in}$ (its transmission) in linear resonant passbands and the red arrows represent the frequency components in the chaotic wave $u_{out}$. In **b**, **d**, the light blue (white) areas represent the band structure of the NAM (the corresponding LAM), where the blue lines are the upper limits of the chaotic bands

the band structure sketched in Fig. 1d, to demonstrate the double-ultra concept based on chaotic bands. The subwavelength meta-cell consists of a Duffing oscillator[51] and a coupled vibro-impact system[52, 53] (Fig. 1c). We expect passbands near LR1 and LR2 to become chaotic bands and reduce wave propagation. A nonlinear meta-cell (Fig. 2a) is achieved by the nonlinear force between permanent magnets and internal collisions. The primary structure is a linear uniform rectangular beam (or square plate) with density $\rho$ and thickness $h$. The lattice constant and width of the beam are $a$ and $b$, respectively. Each attached oscillator consists of three columniform magnets, a columniform strut and a bolt that is used to constrain the magnets. The entire attachment is fixed on the primary beam/ plate at point O. At rest, the magnets are separated from one another by the same clearance $\Delta$. The central magnet with mass $m_r$, is the local resonator in the transverse direction. Other parameters are labeled in Fig. 2.

Nonlinear magnetostatic repulsion forces between the magnet $m_r$ and the other two magnets[54] induce a transverse force $F(x)$ on $m_r$ (see Methods):

$$F(x) \approx k_1x + k_2x^3, \qquad (1)$$

where $x$ is the deviation from the equilibrium position and $-\Delta < x < \Delta$. Therefore, the transverse motion of the attached oscillators can be treated as the Duffing system[51] shown in Fig. 2a where $m_0$ is the equivalent concentrated mass added at point O. Its linear resonant frequency is $f_r$.

As shown in Fig. 2b, the flexural oscillation of the entire attachment is modeled as a linear torsional system $J_O–k_T–J_r$,

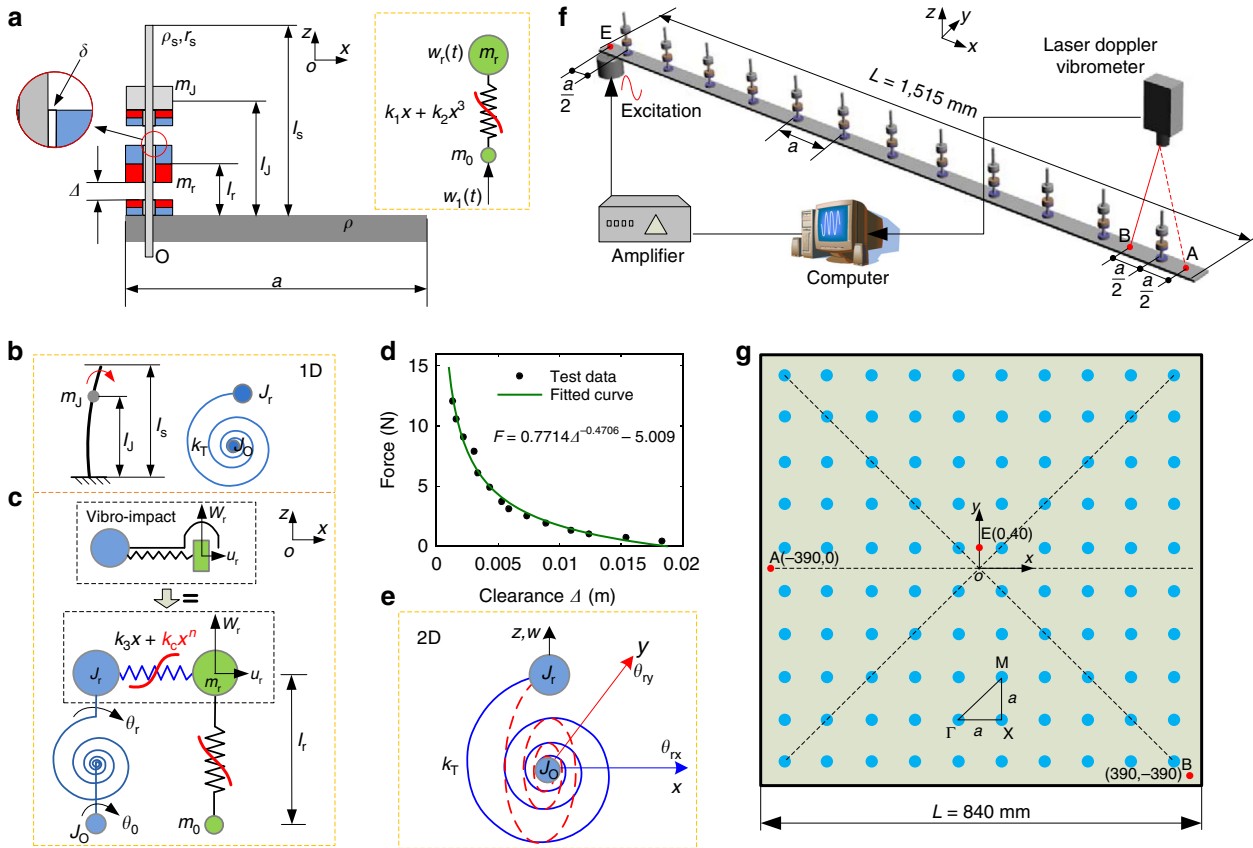

**Fig. 2** Configurations of the NAM beam and plate. **a** Lateral view of the meta-cell with its parameters; the blue and red parts represent three magnets. The gray rectangle with length $a$ is the principle beam/plate, and the columniform strut with length $l_s$ is used to support the magnets. There is a clearance $\delta$ between the strut and the magnet $m_r$. The plot in the dashed box is the equivalent two-degrees-of-freedom (2DoF) Duffing model of the transverse motion of $m_r$. **b** Equivalent torsional motion of the whole attachment in the $xz$ plane. **c** Whole equivalent coupling model for the attachment in NAM beam. **d** Repulsive force–clearance $\Delta$ relation of a pair of magnets. **e** Torsional system in 3D space for the 2D NAM plate. **f** The NAM beam consists of 12 periodic cells and the experimental apparatus. **g** The 2D NAM plate consisting of a square thin plate and $10 \times 10$ periodic attachments represented by blue points. In **f**, **g**, three red points, A, B and E, denote the measurement points, and E is also the excitation point. The positions of these points on the beam are labeled in **f**. For the plate, taking the center point of the square plate as the origin, the coordinates $(x, y)$ of these points are labeled

where $J_O$ is the inertia located at O, $J_r$ is the free moment of inertia and $k_T$ is the stiffness of the torsion spring connecting $J_O$ and $J_r$ (see Methods). As established, $J_r$ also causes a negative index meanwhile the other LR bandgap in the metamaterials is near the natural frequency $f_{T1}$.

Moreover, the magnet features a central hole. A small clearance $\delta = 5 \times 10^{-4}$ m is left between the strut and the magnet $m_r$, and collisions occur in this clearance when the flexural amplitude becomes large. Therefore, the mass $m_r$ generates two different nonlinear interactions during the motions along the transverse and longitudinal directions, respectively. Along the longitudinal direction, it is a vibro-impact oscillator[53] coupled to $J_r$ through the nonlinear force $P(x)$ (see Methods):

$$P(x) = k_3 x + k_c x^n, \qquad (2)$$

where $k_c = \alpha \delta^{-n}, \alpha \approx 1$. The linear part $k_3 x$ derives from the small longitudinal component of $F(x)$. As $\delta << \Delta$, a fair approximation for $k_3$ is $k_3 \approx k_1/10$. Parameter $k_c$ becomes very large as $n$ increases, e.g., $k_c \approx 1 \times 10^{10}$ N m$^{-3}$ for $n = 3$, which indicates that it produces a strong nonlinearity under a smaller amplitude than in the Duffing oscillator. As a compromise, we use $n = 3$ to calculate periodic solutions. A comparative study for larger $n$ is shown in Supplementary Fig. 5. Figure 2c shows the complete equivalent system for the NAM beam. However, for the NAM plate, the torsional motion is equivalent to two identical

coupled vibro-impact systems in 3D space (see Fig. 2e). The structural parameters are listed in Table 1 and other nonlinear factors are neglected in theoretical methods.

A NAM beam and a NAM plate consisting of periodic meta-atoms are shown in Fig. 2f, g and various parameters are listed in Table 2. The experimental methods and apparatuses are described in Methods.

**Double-ultra 1D NAM beam**. The transfer functions, dispersion curves, periodic solutions (i.e., frequency responses) and their bifurcations of the NAM beam are illustrated in Fig. 3. Transfer functions are defined as $H_{A(B)}(\omega) = 20\log_{10}[X_{A(B)}(\omega)/X_E(\omega)]$, where $X(\omega)$ denotes the frequency spectrum and the subscripts A, B and E represent specific measurement points. We compare the dispersion solutions (Fig. 3b, c) and frequency responses (Fig. 3d) of the NAM beams considering only the Duffing oscillator (NAM-N1) and considering both the Duffing and vibro-impact oscillators (NAM-N2) (see Methods).

Four levels of broadband white noise were used to stimulate the NAM beam, as shown in Fig. 3a. From cases i to iv, the nonlinearity strength $\sigma$ (see Methods) increases from 0 (linear) to 4.22 (strongly nonlinear). In both cases i and ii, $\sigma$ is so small that the beam behaves similar to a LAM. The linearized ($k_2 = 0$, $k_c = 0$) dispersion curves for NAM-N1 and NAM-N2 are similar, except that another curve at 10.7 Hz, corresponding to the linearized

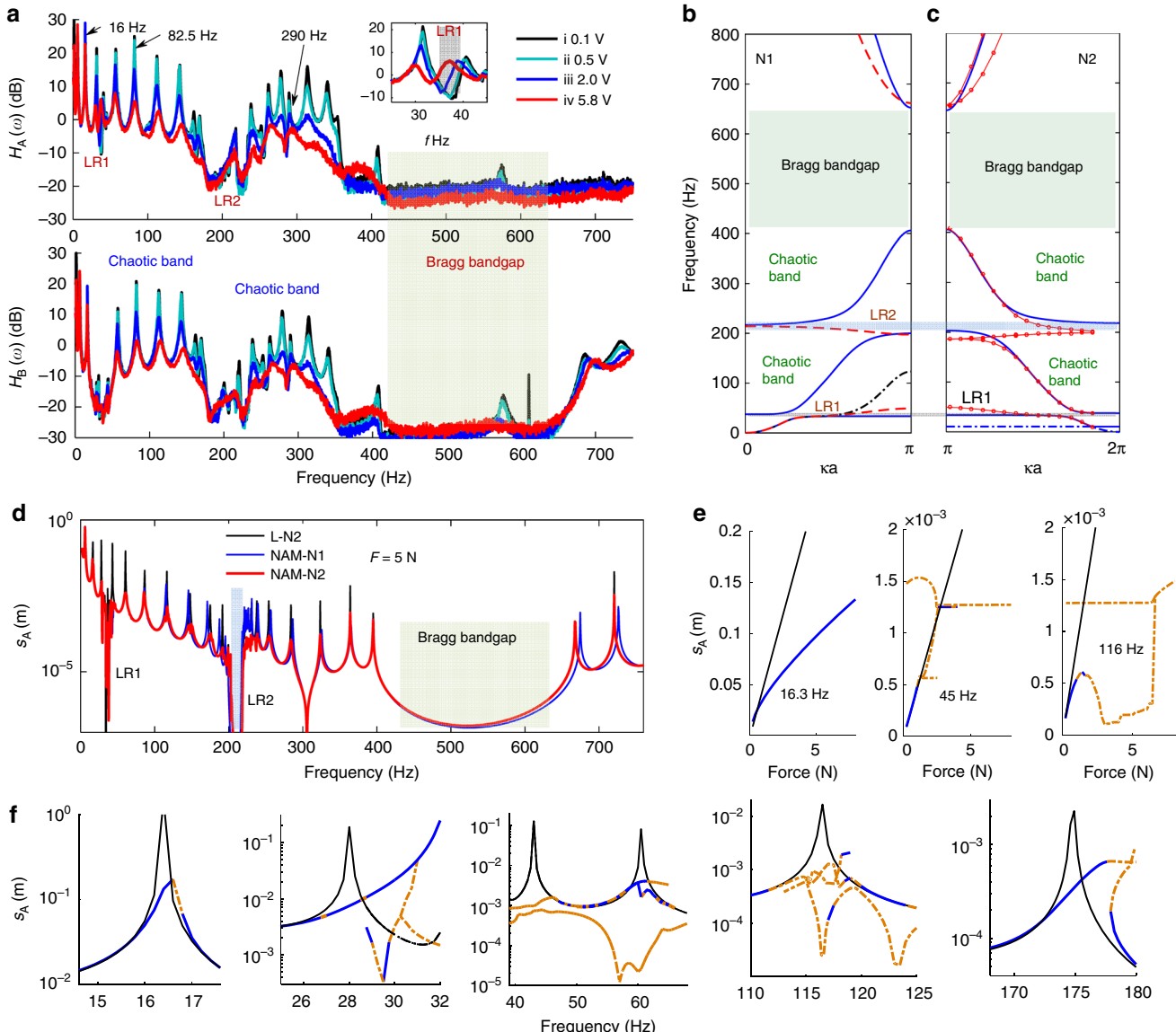

**Fig. 3** Transfer functions, dispersion curves, responses and bifurcations of the NAM beam. **a** Transfer functions of points A and B under four levels of white-noise excitations. The average displacements in the interval (0, 20 Hz) represent the broadband excitation levels; (i: 0.1 V (the voltage of the amplifier), $1.06 \times 10^{-3}$ mm (the average displacement), $\sigma = 7.5 \times 10^{-4} \approx 0$ (the nonlinearity strength)); (ii: 0.5 V, $6.453 \times 10^{-3}$, $\sigma = 0.028$); (iii: 2.0 V, $3.719 \times 10^{-2}$, $\sigma = 0.9262$); and (iv: 5.8 V, $7.936 \times 10^{-2}$, $\sigma = 4.22$). The small plot is the enlarged view near LR1. **b, c** Half dispersion curves of the (**b**) NAM-N1 and (**c**) N2. $\kappa$: the wave vector. In **b, c**, blue lines: LAM; red lines (black dashed line): the NAM solved with the harmonic balance method (HBM) (perturbation approach, PA) with displacement in case iv. In **b**, only the *first curve* of the PA results and *three curves* of the HBM results are shown, because others superpose with the linear results. Three shadings denote the linear bandgaps. **d** Theoretical displacements at point A $s_A$ of the NAM N1, NAM N2 and the linearized LAM in case N2 (L-N2) based on the finite element method (FEM) and HBM (see Methods). In **d, f**, the driving force at point E is $F = 5$ N. **e, f** Bifurcation diagrams under **e** the changing force and **f** the changing frequency. Thin black lines: solutions of LAM; solid blue (*dashed yellow*): stable (unstable) periodic solutions of the NAM

vibro-impact oscillator, appears in N2 (Fig. 3b, c). However, this curve is nearly horizontal, thus no gap opens up. There are two LR bandgaps and a Bragg bandgap below 800 Hz: LR1 near $f_r$ (33.5–37.8 Hz, $\gamma_{lr1} = 0.13$) induced by the linearized Duffing oscillator and LR2 near $f_{T1}$ (200–230 Hz, $\gamma_{lr2} = 0.15$) induced by the torsional motion coupled with the vibro-impact oscillator. The Bragg bandgap in 420–660 Hz ($\gamma_{bg} = 0.57$) is relatively broad. Point B is the node of the flexural modes near $f_r$; therefore, a small $H_B(\omega)$ is obtained in this range. Both $H_A(\omega)$ and $H_B(\omega)$ accurately reproduce the locations of the bandgaps, demonstrating that the theoretical methods are accurate. For the LAM beam, we obtain $\gamma_i < 1$ and $\sum \gamma_i = 0.85$; the passbands are linear resonant

bands in which resonances amplify the waves. The two basic properties make obtaining an ultra-low and broad bandgap to suppress waves in the LAM challenging.

The periodic nonlinear meta-cells create amplitude-dependent properties. In the strongly nonlinear case iv, the nonlinear dispersion solutions for N1 show that the nonlinearity shifts the peak of the first dispersion curve upward; thus, LR1 disappears. Figure 3a demonstrates that enhancing the nonlinearity increases the transmission in LR1 and makes it disappear in case iv. For N2, the curve at 10.7 Hz disappears and LR2 shifts downwards because the high nonlinear stiffness $3k_cx^2$ makes the spring become rigid, which is the limit dispersion solution[48] under a

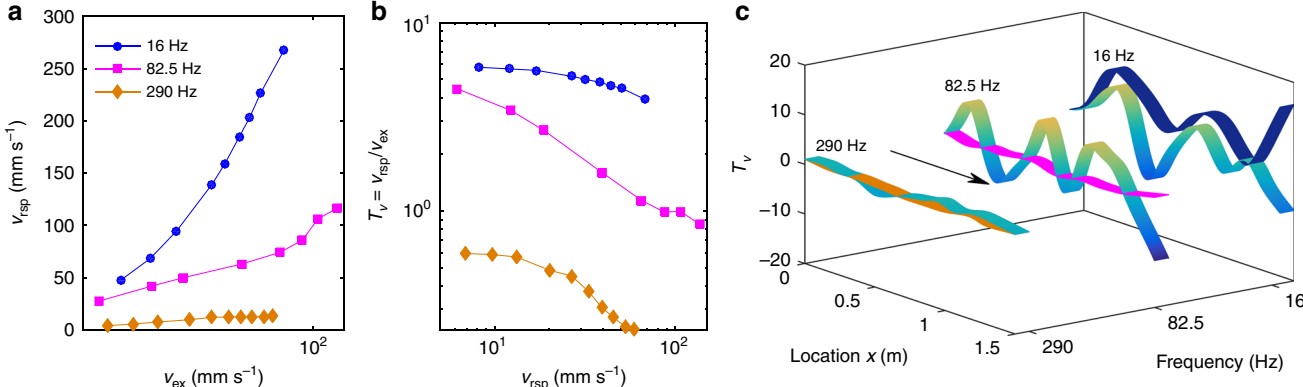

**Fig. 4** Steady responses and wave modes of the NAM beam. **a** $v_{rsp}$ and **b** $T_v$ as a function of $v_{ex}$; the horizontal axes are logarithmic scales. **c** Generalized wave modes of the whole beam under a small excitation (0.3 V, linear regime, shown by the rainbow colors) and a large excitation (5 V, strongly nonlinear regime, shown by the deep blue (16 Hz), magenta (82.5 Hz) and yellow (290 Hz) stripes)

large amplitude. The shifting of LR2 is not noticeable in experiments, because this limit solution overestimates the wave amplitudes, as further demonstrated in Fig. 3d. LR1 and LR2 are termed as nonlinear LR bandgaps. In both N1 and N2, additional curves in LR2 and near the fourth passband of harmonic balance method (HBM) solutions represent unstable waves that do not appear in practice[48], as indicated by $H_{A(B)}$.

Notably, the nonlinearity strongly influences wave propagation in the second and the third passbands, including a small region below LR1. From cases i to iv, relative to the resonant peaks, the wave transmission through the NAM beam decreases by ~20–40 dB in the ultra-low and ultra-broad band of 30–660 Hz (it is ~20 dB in the passbands and it is ~40 dB in the bandgaps). The generalized width reaches $\gamma = 21$, which constitutes a breakthrough compared with conventional LAMs (Supplementary Fig. 7). Therefore, the passbands of the strongly nonlinear AMs can significantly attenuate the elastic waves and enables subwavelength properties with $a = \lambda_f/9$ (see Methods). In the first passband, the influences of the nonlinearity on the three resonances are: significant for 31.56 Hz, moderate for 16 Hz ($H(\omega)$ decreases by 8 dB) and weak for 6 Hz ($H(\omega)$ unchanged), suggesting that the lower the resonance frequency below LR1, the weaker the effect of the nonlinearity[47, 49].

The frequency responses of the LAM and NAM beams confirm the previously detailed properties, as summarized in Fig. 3d (Supplementary Figs. 6, 7). The resonances and bandgaps derived from the theory are in good agreement with the measured $H(\omega)$. However, some discrepancies occur because the parameters and boundary conditions in the theoretical model do not perfectly reproduce the experimental conditions.

A moderate force $F = 5$ N causes the NAM to behave as a strongly nonlinear system (see Fig. 3d). The differences between N1 and N2 illustrate that: the periodic Duffing oscillators are responsible for the wave suppression near LR1 but its influence decreases with increasing distance to LR1 (both below and above LR1); and the vibro-impact oscillators are responsible for wave suppression in the two passbands on both sides of LR2 (Supplementary Note 3). As shown by NAM-N2, LR1 becomes a passband and the resonances in the second and the third passbands are substantially reduced because the linear resonances are replaced by the nonlinear resonances with finite amplitude[47]. The experimental results agree well with the theoretical findings here and that from the discrete models[47–49], supporting the proposed mechanism for nonlinear wave propagation and the band structure of NAMs.

Furthermore, to theoretically demonstrate the chaotic mechanisms of the double-ultra properties and analyze the bifurcations of

periodic solutions, a dimension-reduction algorithm combined with other methods must be adopted (see Methods). As illustrated in Fig. 3e, f, multiple branches are found with the continuation algorithm[49] (Supplementary Note 4). Under a constant force, nonlinear resonances lower than LR1 lead to larger ranges for stable periodic solutions. However, for nonlinear resonances higher than LR1, only unstable branches or alternative stable and unstable solutions are found near the bending peaks. These unstable solutions and alternative properties have been addressed to induce chaos[49]. Under 16.3 Hz (see Fig. 3e), only a monotonous stable branch is found; its amplitude is smaller than the linear solution and the first derivative decreases with increasing force. For the other two cases in Fig. 3e, the nonlinear solutions start along the linear branch and remain stable for a small force; however, then only unstable branches (or a small range of stable solutions), whose amplitudes remain nearly constant, are present, and consequently the transmission decreases. According to these bifurcation properties, it can be predicted that the NAM beam features a quasiperiodic or weakly chaotic response with less ability to reduce the wave transmission near 16.3 Hz. With regard to the waves in the second and the third passbands, they become strongly chaotic and exhibit larger transmission losses[49]. These phenomena are experimentally evidenced in Fig. 4a–c.

To further understand the wave suppression in different bands and to demonstrate the chaotic mechanism observed, we studied the steady responses of the metamaterial at three representative frequencies, 16 Hz, 82.5 Hz and 290 Hz, in the first, the second and the third passbands, respectively. All are near (but not coincident with) the linear eigenfrequencies (see Fig. 3a). In Fig. 4, the transmission $T_v = v_{rsp}/v_{ex}$, where $v_{rsp}$ and $v_{ex}$ represent the velocity amplitudes at points A and E, respectively. In contrast to linear cases, here $T_v$ rapidly decreases to a value less than 1, whereas $v_{rsp}$ increases with the increasing $v_{ex}$. This behavior demonstrates that a stronger nonlinearity corresponds to a larger transmission loss[47] in the studied nonlinear range. Furthermore, $T_v$ at 82.5 Hz and 290 Hz decrease more than at 16 Hz, in good agreement with the results in Fig. 3. In Fig. 4c, these wave reduction and suppression effects are further described by the generalized wave fields along the beam.

**Double-ultra 2D NAM plate.** The results of the scanning experiments (see Methods) on the 2D NAM plate are shown in Fig. 5a, b. As with the beam, from cases i to iv, the average driving displacement increases by 22 times and $\sigma$ increases from $\sigma \approx 0$

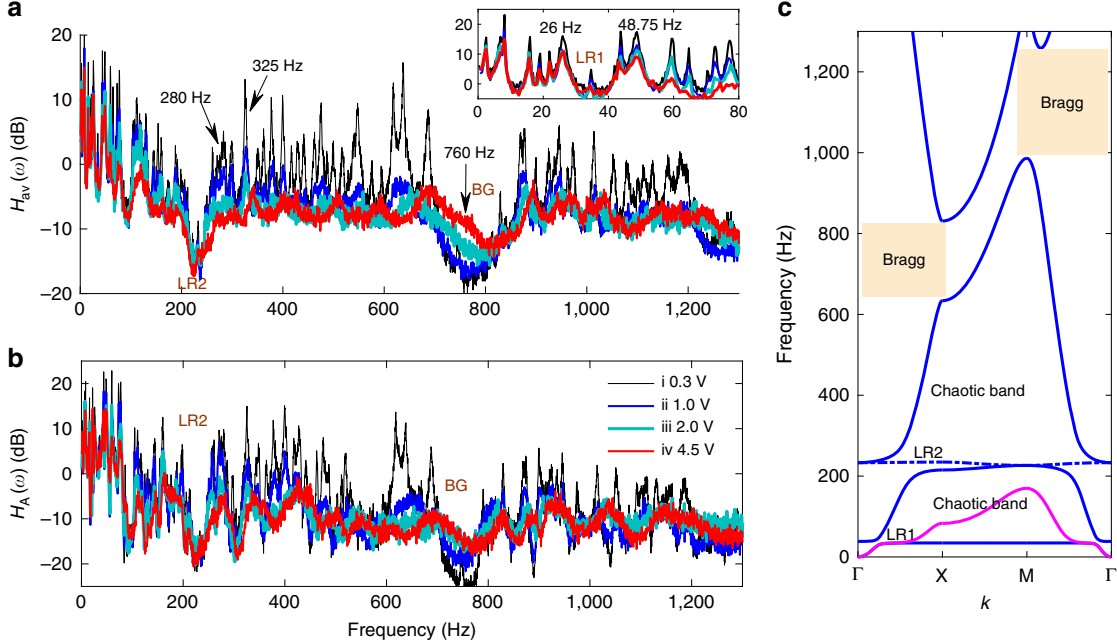

**Fig. 5** Transfer functions and dispersion of the NAM plate. Excitations are broadband white noises. Four driving levels are (i: 0.3 V, 3.2578 × 10⁻³ mm, $\sigma = 0.0067 \approx 0$); (ii: 1.0 V, 1.5748 × 10⁻², $\sigma = 0.157$); (iii: 2.0 V, 3.2509 × 10⁻², $\sigma = 0.669$); (iv: 4.5 V, 7.1242 × 10⁻², $\sigma = 3.21$). **a** Average transfer function $H_{av}(\omega)$ of the 13 × 13 scanning points; the iconograph is the enlarged view near LR1. **b** Transfer functions of point A, $H_A(\omega)$. BG denotes Bragg bandgap. **c** Dispersion curves of the 2D metamaterial plate considering only the Duffing oscillator; blue curves: LAM; and magenta line: NAM solved with perturbation approach with displacement amplitude in case iii, other curves are superposed with blue curves (see Methods). $k$ denotes the wave vector of the first Brillouin zone (see Fig. 2g)

(linear) to $\sigma = 3.21$ (strongly nonlinear). The average transfer function is $H_{av}(\omega) = \sum_{i=1}^{N} H_i(\omega)/N$, where $N = 169$.

As shown in Fig. 5c, two omnidirectional LR bandgaps open up near $f_{lr1} = 34.65$ Hz (34.5–38 Hz, $\gamma_{lr1} = 0.1$) and $f_{lr2} = 230$ Hz (216–245 Hz, $\gamma_{lr2} = 0.14$) because of the transverse motion of $m_r$ and the coupled vibro-impact system, respectively. The dashed line in LR2 corresponds to the LR mode of the described torsional motion. Both LR1 and LR2 are nonlinear bandgaps. In addition, two directional Bragg bandgaps exist along ΓX and ΓM in the interval 635–835 Hz ($\gamma_{\Gamma X} = 0.32$) and near 1,150 Hz, respectively. However, $\gamma_i << 1$ and $\sum \gamma_i = 0.56$; thereby, they are narrow. The LR bandgaps are clearly visible in $H_{av}(\omega)$ and $H_A(\omega)$, as displayed in Fig. 5a, b. The directional bandgap along ΓX also helps suppressing the average response. Because of the dense flexural modes in this 2D LAM, the passbands feature dense resonances that enhance the incident waves.

For the 2D NAM plate, the perturbation result shows that the first dispersion curve gets significantly distorted because of the occurrence of the strong nonlinearity. In contrast to the linear case i, the strengthened nonlinearity suppresses the broadband resonances. In fact, a moderate nonlinearity in case ii enables suppression of the resonances between 30 and 1,200 Hz, especially in the range 200–1,200 Hz. Further enhancing the nonlinearity, as in case iv, leads to reductions both of $H_{av}(\omega)$ by 20–40 dB in the range 50–1200 Hz and of the resonances by 10 dB in the range 30–50 Hz. We obtain a subwavelength property with $a = \lambda_f/10$ at 30 Hz here. Resonances in the first passband are minimally reduced. This behavior is the same as that observed in the NAM beam. Therefore, between 30 and 1,200 Hz this 2D NAM features the double-ultra property that the wave transmission is significantly reduced. The generalized width $\gamma = 39$ is nearly double the width of the NAM beam.

To further describe the double-ultra properties, we measured the steady responses at points A and B after a monochromatic excitation at point E (see Fig. 6). As expected, the 2D nonlinear

flexural mode depends on both the frequency and the position, and the transmission does not vary monotonically. At 26 Hz (Fig. 6a, e), although $v_{rsp}$ increases with the increasing driving amplitude, $T_v$ reaches the maximum value at $v_{ex} = 0.8$ mm s⁻¹ but then decreases by a factor of 2.7 at $v_{ex} = 9$. A similar behavior is observed at 48.75 Hz (Fig. 6b, f). In contrast, $v_{rsp}$ at 280 Hz (Fig. 6c, g) and 325 Hz (Fig. 6d, h) first increase monotonically against $v_{ex}$ and then remain nearly constant, while the transmission decreases substantially. Moreover, in the four cases except for a small region near $v_{ex} = 0.9$ mm s⁻¹ at 325 Hz, the amplitude and transmission at A and B vary synchronously.

As shown in Fig. 5a, b, $H(\omega)$ along ΓX for frequencies in the directional bandgap near 760 Hz increases with $\sigma$. Different experiments were conducted at 760 Hz to illustrate this effect. As shown in Fig. 7a, when increasing $v_{ex}$ from zero, the waves are first suppressed so the responses vary along a low-energy orbit corresponding to the bounded state. The responses then jump up to a high-energy orbit at a critical point, that is, the excited state. With decreasing excitation, the responses jump down to the bounded state at a smaller critical point: a hysteresis loop is observed between the two states. The jump is relevant to saddle-node bifurcations[49, 55]. In Fig. 7b, the contour plot of $\Delta T_v$ illustrates that elastic energy is transferred from the central area to the boundaries in the excited state, which breaks the bandgap effect. Therefore, the bandgaps in NAMs feature multi-state behavior switching from one state to the other by jumps[49]. High-dimensional acoustic devices based on such behavior are conceivable.

**Confirmation of chaotic waves in experiments.** The double-ultra properties are relevant to the propagation states of the waves: periodic, quasiperiodic or chaotic. To understand the transition between periodic and chaotic states, and to further clarify the chaotic features and band structures, we analyzed the

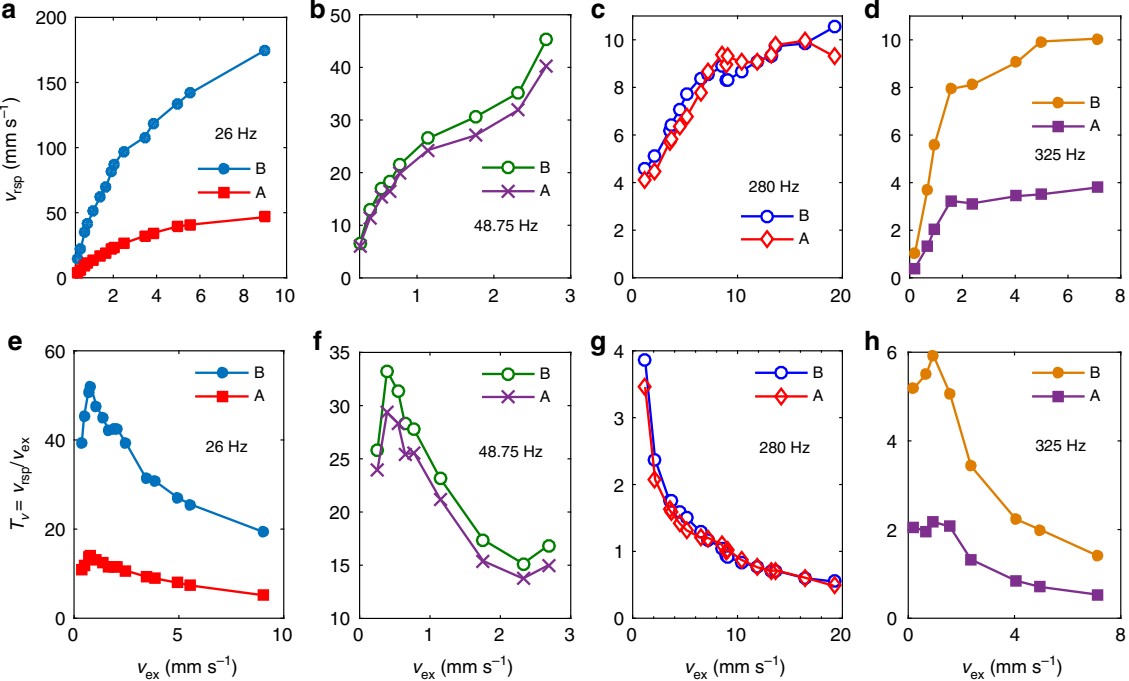

**Fig. 6** Steady responses of the NAM plate. **a–d** Steady response amplitudes $v_{rsp}$ and **e–h** the corresponding transmissions $T_v$ at points A and B changing with $v_{ex}$ under four frequencies 26, 48.75, 280 and 325 Hz in the first, the second and the third passbands, as labeled in Fig. 5a. In each legend, the frequency indicates that the diagram is relevant to this frequency and A (B) denotes $v_{rsp}$ or $T_v$ at point A (B) under that frequency

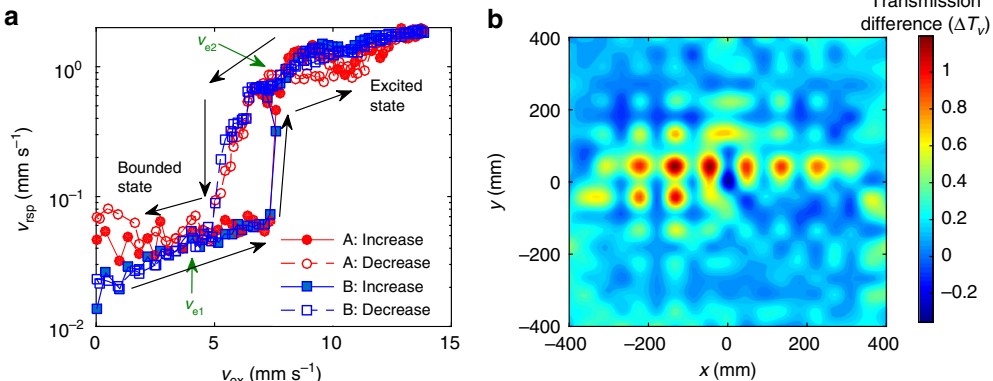

**Fig. 7** Multi-states in the bandgap of NAM plate. Frequency is 760 Hz. **a** Response amplitudes $v_{rsp}$ at points A and B varying against the linearly and continuously varied driving amplitude $v_{ex}$ in time domain, i.e., the varying-amplitude experiment (see Methods). In the legend, Increase (decrease) refers to the process increasing (decreasing) $v_{ex}$. Black arrows show the varying routes. **b** Contour plot of the transmission difference $\Delta T_v$ between $v_{e1} = 3.13$ and $v_{e2} = 8.55$ labeled in **a**, $\Delta T_v = T_{v2} - T_{v1}$. Scanning experiments were carried out to obtain this plot

spectra and the largest Lyapunov exponents (LLEs, $\lambda_m$)[50, 56–58] (see Methods) of the steady waves.

Theoretical investigations[47–49] have revealed that the resonances are suppressed by the chaos induced by periodic-doubling bifurcations. The power spectra $P_A(\omega)$ at 290 Hz in the NAM beam (see Fig. 8a) clearly illustrate the period-doubling route to chaos. In the linear regime ($v_{rsp} = 3.137$ mm s$^{-1}$), the energy is localized at the driving frequency $f_d$; increasing $v_{rsp}$ to 9.77 mm s$^{-1}$ generates period-doubling frequencies that divides the elastic energy; further increasing the amplitude redistributes the wave energy in a broad, higher band, which is chaotic. This phenomenon has been termed energy dispersion[47].

First, we analyzed the LLEs of the NAM beam at the three frequencies in Fig. 4. As shown in Fig. 8b, LLE of the driving velocity fluctuates near 0 (see Methods) over a large amplitude

range. The maximum value <0.05 is positive but small, ensuring that noises in the driving forces have a negligible influence, even if the forces are large. By considering the errors, $\lambda_{mc} = 0.05$ is chosen as the critical value of LLE where switching from the quasi-period to chaos occurs. In Fig. 8b, c, $\lambda_m$ fluctuates in a non-monotonic way but still exhibits an increasing trend on the whole; when $v_{ex}$ is very small, $\lambda_m < \lambda_{mc}$, indicating that the responses are periodic (or quasiperiodic). Further increasing $v_{ex}$ causes $\lambda_m$ to rapidly pass though zero and $\lambda_{mc}$ to satisfy $\lambda_{mc} \leq \lambda_m < \infty$, which indicates that the waves become chaotic. In the chaotic regime $\lambda_m \gg 0$, at 82.5 and 290 Hz, which implies that chaotic behaviors are strong and that the trajectories in the chaotic attractors quickly become separated. By contrast, $\lambda_m < 0.1$ at 16 Hz denotes the weak chaos that is approximate to a quasiperiodic orbit; therefore, the material has a weaker ability

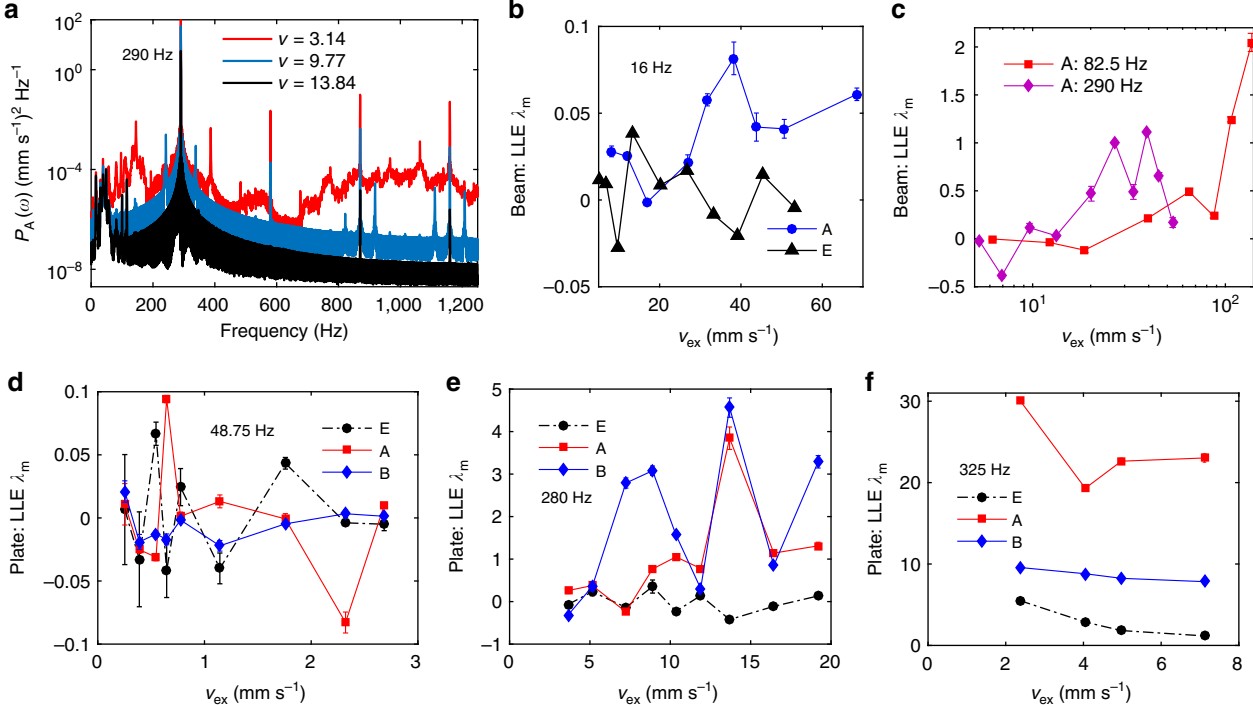

**Fig. 8** Chaos and state transition in the NAMs. These signals under different frequencies are steady responses and excitations shown in Figs. 4, 6 for the beam and plate, respectively. **a** Power spectra $P_A(\omega)$ of point A under 290 Hz of the beam, the legend shows different response amplitudes. **b–f** Error graphs of the change of signal LLEs $\lambda_m$ with $v_{ex}$, where the lengths of the bars indicate the computational errors; **b–c** for the beam and **d–f** for the plate. In the legends of **b–f**, A, B and E represent the signals at the corresponding measured points; every labeled frequency indicates $\lambda_m$ under that frequency

to suppress the waves in frequency ranges below LR1[49]. These results confirm the predictions made after the bifurcation analyses in Fig. 3e, f.

Next, we calculated $\lambda_m$ of the NAM plate under the three frequencies in Fig. 6. The results obtained at 26 Hz are similar to those obtained at 48.75 Hz; therefore, only the latter are presented. At 48.75 Hz (see Fig. 8d), $\lambda_m$ fluctuates near 0 and $\lambda_m < 0.1$ in the explored amplitude range at the three points E, A and B. Therefore, we deduce that low-frequency waves are periodic, quasiperiodic or weakly chaotic. At 280 Hz (Fig. 8e), $\lambda_m$ at point E still fluctuates near 0; in contrast, $\lambda_m$ at A and B increases to 4.5 so that strong chaos occurs, which suppresses the waves, and a periodic window appears at $v_{ex} = 11.89$. In the case of 325 Hz (Fig. 8f), the signal undergoes low-frequency noise therefore $\lambda_m$ at point E has a large value. Fortunately, $\lambda_m$ at points A and B are much larger, and $\lambda_m > 20$ at A, which corresponds to a strong chaos. Moreover, although the amplitudes measured at A and B are approximately equal (see Fig. 6), their LLEs differ substantially in some intervals, which indicates that although the amplitudes may vary synchronously at different points, one response may be strongly chaotic while the other is periodic.

The aforementioned quantitative statements experimentally establish that the passbands of the NAMs become chaotic, which agrees with the bifurcation analysis. Moreover, the regularities of the LLEs are mutually consistent with the theoretical findings outlined in the discrete model[49]. Therefore, the experiments demonstrate the chaotic mechanism and the features of chaos in the NAMs.

## Discussion

We designed a NAM beam and a plate with a strongly nonlinear meta-cell consisting of a Duffing oscillator and a linear torsional oscillator coupled to a vibro-impact oscillator. Our proof-of-

concept experimental results with the NAMs demonstrated that these metamaterials overcome the bandwidth limit ($\gamma < 1$) of conventional LAMs by at least two orders of magnitude: with subwavelength ($\sim \lambda_f/10$) cells, the generalized width reaches an exciting value $\gamma = 21$ in the 1D NAM and it increases to $\gamma = 39$ in the 2D NAM. In these broad bands (which consist of bandgaps and chaotic bands), the transmission of waves is reduced by as much as 20–40 dB. We demonstrated both theoretically and experimentally that the double-ultra effect is attributable to strong chaos, because the propagating elastic waves become chaotic under periodic incidents. The band structures and features of chaos are also consistent with the theoretical results. Moreover, bandgaps in the NAM exhibit a multi-state behavior; switching from one state to the other arises by jumps.

In conclusion, this study demonstrates the chaotic band in NAMs, which can significantly reduce the wave transmission in an ultra-low and ultra-broad band. Our work unveils the physical effects of NAMs and enables further progress in understanding NAMs. We envision that the chaotic band induced double-ultra wave suppression will open new opportunities for vibration and noise control, acoustic energy transfer and dissipation and elastic wave manipulation. The multi-state behavior presents an advantage in controlling the state of acoustic devices.

## Methods
**Metamaterial samples**. The parameters of the NAM samples shown in Fig. 2 are listed in Table 1. The permanent magnet is made of the neodymium–iron–boron alloy. Its outer diameter, inner diameter and the thickness are 15, 5 and 4 mm, respectively. The oscillator $m_r$ consists of two magnets. A magnet is fixed on both the beam and the bolt, respectively, that is the installation scheme of the magnets is one–two–one. In Fig. 2a, $\rho_s$ ($r_s$) denotes the density (the radius) of the strut; $m_J$ denotes the total mass of the upper magnet and its bolt.

The flexural wavelength of the primary structure is

$$\lambda_f = 2\pi (EI/\rho bh)^{1/4} \omega^{-1/2}, \qquad (3)$$

## Table 1 Parameters of the NAMs

| Para. | $a$ (mm) | $b$ (mm) | $h$ (mm) | $m_O$ (g) | $m_r$ (g) | $m_J$ (g) |
|---|---|---|---|---|---|---|
| 1D | 120 | 20 | 3.8 | 15 | 10 | 9 |
| 2D | 80 | – | 2 | 8 | 10 | 9 |
| Para. | $\Delta$ (mm) | $l_r$ (mm) | $l_J$ (mm) | $l_s$ (mm) | $r_s$ (mm) | $\lambda_f$ (m) |
| 1D | 12.5 | 20.5 | 45 | 70 | 1.9 | 1.074 |
| 2D | 12 | 20 | 40 | 70 | 1.9 | 0.805 |

Here $\lambda_f$ denotes the flexural wavelength at 30 Hz

## Table 2 Equivalent parameters of the attached structure

| Para. | $k_1$(N m$^{-1}$) | $k_2$(N m$^{-3}$) | $J_O$ (kg m$^2$) | $J_r$(kg m$^2$) | $k_T$(N m rad$^{-1}$) |
|---|---|---|---|---|---|
| 1D | 448.67 | 3.276e6 | 5.750e-7 | 2.225e-5 | 44.874 |
| 2D | 473.92 | 2.93e6 | 5.820e-7 | 1.838e-5 | 49.045 |

where $E$ is elastic modulus and $I = bh^3/12$. The flexural wavelength in the pure plate is

$$\lambda_f = 2\pi(D_0/\rho h)^{1/4}\omega^{-1/2}, \tag{4}$$

where $\omega$ denotes the angle frequencies; $D_0 = Eh^2/12(1-\mu^2)$; $\rho$ and $\mu$ are the density and the Poisson's ratio of the material, respectively. The beam, the plate and the struts are all made of aluminum. It's $E = 70$ GPa, $\mu = 0.3$ and $\rho = 2,780$ kg m$^{-3}$.

**Equivalent motions**. The magnetostatic repulsion force between the permanent magnets can be expressed as $F(\Delta) = C\cdot\Delta^{-p} + C_0$, $p > 0$, where $C$ and $C_0$ are constants. $C_0$ is introduced to better fit the measured data and in theory $C_0 = 0$. Therefore, the transverse force (along $z$ axis) on $m_r$ reads[54] $F(x) = C[(\Delta - x)^{-p} - (\Delta + x)^{-p}] \approx k_1 x + k_2 x^3$, i.e., Eq. (1). We measured the nonlinear repulsive force-clearance relation $F(\Delta)$ between two identical magnets, as illustrated in Fig. 2d. The stiffness coefficients $k_1$ and $k_2$ derived from the measurements are listed in Table 2. Because of the cubic nonlinear term $k_2 x^3$ in Eq. (1), the transverse attached oscillators can be treated as the Duffing system[51] represented in Fig. 2a. Its transverse motion equations read

$$m_O\ddot{w}_1 = F_1(t) + k_1(w_r - w_1) + k_2(w_r - w_1)^3, \tag{5}$$

$$m_r\ddot{w}_r = -k_1(w_r - w_1) - k_2(w_r - w_1)^3, \tag{6}$$

where $w_1$ and $w_r$ are transverse displacements of $m_O$ and $m_r$, respectively; $F_1(t)$ is the node force applied on $m_O$, which is generated by the shearing stress in the beam or plate; the double overdot denotes a second-order time derivative. The linearized natural frequency of this Duffing oscillator is $f_r \approx 35$ Hz.

Moreover, the bending moment causes the entire attachment undergoing flexural oscillations (see Fig. 2b). This part is modeled as a small beam attached with a concentrated masse $m_J$ at the location $l_J$, which induces a considerable moment of inertia. Therefore an entire attachment in low frequency is equivalent to the linear torsional system $J_O$–$k_T$–$J_r$.

In fact, the collision occurs in the small clearance $\delta$ when the torsional amplitude is not small, which provides the other strongly nonlinear source in our NAMs. The elastic impact interaction force follows the power law[53] $\alpha(x/\delta)^n = k_c x^n$, $k_c = \alpha\delta^{-n}$, where $n > 1$ is an odd number and $\alpha$ depends on the estimated peak force (or acceleration) in the collision. The value of $\delta^{-n}$ is so large that the influence of $\alpha$ becomes weak, here $n \approx 1$. The function becomes rectangular if $n \to \infty$. Therefore, the motion of $m_r$ along the longitudinal direction is a vibro-impact oscillator that couples with $J_r$ through the nonlinear force $P(x)$ in Eq. (2). For the NAM beam, the torsion motion of the entire attachment occurs in the $xz$ plane only and the complete equivalent system of a cell is shown in Fig. 2c. The motion equations of this coupling nonlinear system are

$$J_O\ddot{\theta}_O = k_T(\theta_r - \theta_O) + M_O(t), \tag{7}$$

$$J_r\ddot{\theta}_r + m_r l_r \ddot{u}_r = -k_T(\theta_r - \theta_O), \tag{8}$$

$$m_r\ddot{u}_r = -k_3(u_r - l_r\theta_r) - k_c(u_r - l_r\theta_r)^n, \tag{9}$$

where $\theta_O$ and $\theta$ are torsional angles of $J_O$ and $J_r$, respectively; $M_O(t)$ is the bending moment generated by the primary beam and $u_r$ is the longitudinal displacement of

$m_r$. Other nonlinear factors are analyzed and can be neglected (see Supplementary Figs. 4, 6).

For the uncoupled 2DoF torsional system, by equations $\ddot{\theta}_O = -\omega^2\theta_O$, $\ddot{\theta} = -\omega^2\theta$, we can solve $\theta$ in terms of $\theta_O$, so that, in terms of the angle $\theta_O$, one yields

$$M_O = \bar{J}(\omega)\ddot{\theta}_O, \tag{10}$$

$$\bar{J}(\omega) = J_O + k_T/(\omega_{0T}^2 - \omega^2), \tag{11}$$

where $\omega_{0T}^2 = k_T/J_r$ and $\bar{J}(\omega)$ symbolizes the equivalent dynamic inertia of the whole attachment at frequency $\omega$. This expression is similar to the LR oscillator in an AM[3]. $\bar{J}(\omega)$ indicates that the moment of inertia can also generate a negative index and introduces another LR bandgap in the metamaterial near the natural frequency $f_{T1} = \omega_{0T}/2\pi$. The three equivalent parameters, $J_O$, $J_r$ and $k_T$, can be determined based on the FEM[59], as detailed in Supplementary Note 1. Their values are listed in Table 2, and $f_{T1} = 226$ Hz (Supplementary Fig. 1).

In the NAM plate, the bending moment drives the attachment to generate flexure motion in the 3D space (see Fig. 2e) but not in a 2D plane as the AM beam does. Therefore, the equivalent system has six DoFs and the motion equation of this part reads

$$J_O\ddot{\theta}_{Oi} = k_T(\theta_{ri} - \theta_{Oi}) + M_{Oi}(t), \tag{12}$$

$$J_r\ddot{\theta}_{ri} + m_r l_r \ddot{u}_{ri} = -k_T(\theta_{ri} - \theta_{Oi}), \tag{13}$$

$$m_r\ddot{u}_{ri} = -k_3(u_{ri} - l_r\theta_{ri}) - k_c(u_{ri} - l_r\theta_{ri})^n, \tag{14}$$

where $\theta$ and $M$ denote the torsional angle and bending moment, respectively; $i = x, y$; subscripts $x$ and $y$ symbolize the coordinates $x$ and $y$; the subscripts r (O) represents the variables of the resonator (the fix point O on the plate). As the two systems are identical, by considering the oscillation in the $xz$ plane only, we can obtain the equivalent parameters, which is same with the 1D beam. The results are listed in Table 2, and $f_{T1} = 260$ Hz, $f_{T2} = 1,484$ Hz.

The strength of nonlinearity is defined by

$$\sigma = nk_c A^{n-1}/k_3, \tag{15}$$

where $A$ stands for the response amplitude used to estimate the nonlinear strength[47, 49]. $\sigma$ is a relative indicator. A larger $\sigma$ is the stronger is the nonlinearity.

**Experiment apparatuses and measurements**. The experimental configuration and testing scheme for the NAM beam and plate are shown in Fig. 2. The metamaterial beam and plate were fixed to an electromagnetic exciter at point E. The output driving velocities and displacements were adjusted by modulating the voltage of the amplifier.

For the NAM beam, the other end was free; two test points on the primary beam, A and B, were set on the symmetrical sides of the 12th attachment (see Fig. 2f). Two types of experiments were implemented with the NAM beam: broadband frequency responses and responses under monochromatic excitations. For the broadband responses, the random broadband white noise acted as the driving force and the response velocities at points A, B and E, were directly measured. In the other experiment, monochromatic sinusoidal waves were employed to drive the beam, and the responses in the time domain at points A and E were measured. In both experiments, the response velocities at the three points were measured synchronously using three laser Doppler vibrometers. In addition, for the measurement of the wave shapes of the whole beam, a vibrometer was used to scan points along the primary beam.

For the NAM plate, the excitation point E was near but not coincident with the center point of the plate. Other boundaries of the plate were free. This driving method can excite the non-symmetrical modes of flexural waves in the plate. The positions of measurement for points A, B and E are shown in Fig. 2g. The two types of experiments were also implemented on the 2D NAM plate. However, in the broadband frequency responses experiment, $13 \times 13$ scanning points were set on the primary plate, and a vibrometer was used to measure their velocities one by one. In addition, we used a varying-amplitude monochromatic experiment to measure multiple states in the bandgap. Here the driving amplitude was $A(t)\sin(2\pi f_d t)$, where $f_d$ is constant but the amplitude $A(t)$ changes linearly with time. Moreover, to measure the wave fields in the bounded state and excited state under 760 Hz, $19 \times 19$ scanning points were set on the primary plate. In the scanning experiments, the measured points were distributed uniformly in a square region from the coordinate (−400, 400) to (400, −400).

**Signal processing**. Nonlinear time series analysis methods were employed to analyze signals under monochromatic excitations. Under periodic incidents, the propagated waves in the NAM beam follow periodic, quasiperiodic or chaotic trajectories[51]. The chaotic trajectories in its attractor diverge, on average, at an exponential rate over the time evolution characterized by the largest Lyapunov exponent (LLE) $\lambda_m$[56]. Therefore, the LLE can quantitatively identify whether the

signal is chaotic. In theory, if $\lambda_m < 0$, the motion is periodic having a stable fixed point; if $\lambda_m = 0$, it is a quasiperiodic response having a stable limit cycle; and if $0 < \lambda_m < \infty$, it is chaotic. However, errors and noises from the algorithm and experiment make using a single value $\lambda_m = 0$ to identify the quasiperiodic state in experiments almost impossible.

We employed an algorithm for the LLE derived by Kantz et al.[56–58] because of its robustness to the noise; this algorithm is based on the phase space reconstruction technique. The proper embedding dimension $d_E$ and time delay $\tau$ should be determined to reconstruct the phase space[56]; the autocorrelation function was calculated to determine the optimal lag $\tau$, and the false nearest neighbors method was used to determine the optimal $d_E$.

**Dispersion theories.** We established the finite element models of the meta-cells of both the NAM beam and plate. For NAM-N1, perturbation approach (PA) was used to calculate the approximate dispersion solutions. For NAM-N2, only the HBM could be used and the analytical solutions of the dispersion equations were solved. However, for the NAM plate-N2, the high dimensions of equations made finding analytical solutions difficult. Therefore, only the dispersion solution solved by PA of the NAM plate N1 is presented (see Supplementary Note 2 for more details).

**Periodic solutions and bifurcations.** A periodic solution is also a steady frequency response. To calculate the frequency responses, we used a standard finite element procedure[59] to obtain the motion equations for the whole NAM beam. Then, we used HBM to solve the approximate solutions. The numerical Newton method helps finding the solutions (Supplementary Note 3).

In the case NAM beam-N2, the finite element model has 90 dimensions. However, to analyze the stabilities and bifurcations of the periodic solutions, we need to reduce the dimensions of the whole model. We adopted the post-processed Galerkin algorithm in the frequency domain to reduce the dimensions of the NAM beam-N2 (Supplementary Note 4). A picking dimension-reduction procedure reduced the dimensions from 90 to 23. With the reduced system, we analyzed the periodic solutions and their bifurcations using the harmonic average method (Supplementary Figs. 8, 9).

**Data availability.** The experimental data that support the findings of this study are available in Dryad Digital Repository (http://datadryad.org/) with the identifier DOI:10.5061/dryad.6m8nt[60].

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

## Acknowledgements

This research was funded by the National Natural Science Foundation of China (Projects Number 51405502, number 51275519 and number 11372346).

## Author contributions

J.W. supervised the experiments; X.F. proposed the idea and designed the experimental proposal; X.F., J.W., J.Y. and D.Y. prepared the experimental set-up; X.F. performed the experiments; X.F. and B.B. carried out the theoretical calculations; X.F., J.W. and B.B. wrote the paper; all authors analyzed the data, discussed the results and commented on the manuscript.

## Additional information

**Competing interests:** The authors declare no competing financial interests.

