## [Peer Review File · Nature Communications]

Reviewers' comments:

Reviewer #1 (Remarks to the Author):

The paper study the nonlinear acoustic metamaterials with Ultra-low- and ultra-broad-band. The mechanism has not explained clearly. Please describe the ultra-low and ultra-broad band results and give the physical reason.

Here are some questions in details:

1. What the meaning of γ , is it generalized width or width-position ratio?
2. What is the meaning $a \approx 2\lambda/5$?
3. What is the meaning of m_0 in Figure 1?
4. Conceptual diagram of the ultra-low- and ultra-broad-band mechanism using the chaotic band

Question: How to realize the conceptual diagram using nonlinear acoustic metamaterials?

5. - chaotic band-reduce the wave transmission by as much as 20-40 dB

Question: The receiving signal is very small, it may influenced by noise?

6. By contrast, the essence of the ultra-low- and ultra-broad-band (double ultra) NAM is enabled by the chaotic responses coming from the unique strongly nonlinear meta-cells.

Question: The chaotic response is hard to control?

7. The nonlinear effect is broadband and it is independent of the width of the bandgap in NAM but it depends on the location of the nonlinear LR bandgap.

Question: Why the nonlinear effect is broadband and it is independent of the width of the bandgap in NAM? And the nonlinear effect is small

8. What is the difference between the work and the references 47-49?

9. The experiments is proof-of-concept experimental. How to use it in practice?

Reviewer #2 (Remarks to the Author):

The paper presents new and interesting information. It can be published after introducing multiple corrections. I may suggest the following corrections:

1. Using unexplained symbols in the Abstract should be discouraged.
2. Some statements are not accurate and in fact they are misleading. For example, in page 1 authors write "In the nonlinear electromagnetic and photonic metamaterials [23,24], desired nonlinear responses [25,26] and many unique properties have been found and demonstrated [27-31]. However, nonlinear acoustic metamaterial (NAM) is a new topic, and their physical properties are waiting to be unveiled". But the situation is quite opposite. As a rule nonlinearity in electromagnetic and photonic metamaterials is weak unlike strong nonlinearity (and even nonlinearizable nonlinearity) was observed in acoustic metamaterials, e.g. granular crystals described in [37, Chapter 1] and in many subsequent publications.
3. Some references incorrectly represent the state of the art. For example, in page 1, the

statement "The presence of nonlinear media in linear periodic structure can be used to realize ultrasound acoustic diodes [32,33]" ignores the fact that acoustic diode was presented in the paper V.F. Nesterenko, et al., Physical Review Letters, vol. 95, pp. 158702, 2005 which was published four and five years before papers [32,33]. This fact is clearly confirmed by the citation from this PRL2005 paper "In summary we observed a strong sensitivity to the initial precompression of the reflected and transmitted energy from the interface of the two granular media. This phenomenon can be named the acoustic diode effect. It can be employed for designing tunable information transportation lines with the unique possibility to manipulate the signals delay and reflection at will, and decompositions/scrambling of security-related information".

4. In the text authors use ' $\mu_{b,b}$ ' but in the Fig. 1 they use ' $\mu_{b,0}$ ' which was not introduced in the text.

5. In the page 6 authors should clearly explain that expansion related to Eq. 1 is valid when amplitude of displacement is much smaller than the clearance (and not just smaller as in the text).

6. The sentence in the page 9 "Further increasing the driving amplitude strengthens the nonlinearity to be non-negligible" is awkward. Probable replacement would be "Further increasing the driving amplitude strengthens the nonlinearity which can't be neglected".

7. The statement in page 10 "The responses of the NAM are amplitude-independent in practice" is confusing and makes the subject of the whole paper questionable. Indeed the focus of the paper is on the nonlinear acoustic metamaterials and nonlinearity is supposed to make responses of NAM amplitude dependent!

8. Authors use terms "strongly nonlinear system" (page 10), "stronger nonlinearity", "different nonlinear strengths" (page 12), "strong nonlinearity" (page 14), "unique strongly nonlinear met-cells", "strong nonlinearity" (page 19). It is not acceptable unless authors attribute some numbers to these terms. The possible suggestion would be the characteristic values of the ratio of two terms (linear and nonlinear) in Eq. 1 for corresponding cases.

9. Finally, the English in the paper should be polished.

Reviewer #3 (Remarks to the Author):

This paper performs novel analytical and experimental analyses to uncover chaotic bands in one-dimensional (beams with attached nonlinear resonators) and two-dimensional (plates with similar attachments) systems. The systems considered, incorporating magnetically-levitated masses, are a clever means for achieving the desired nonlinearity. The analysis of this system is rigorous and thorough, and convincingly captures the system dynamics. Subsequent studies of the wave propagation as a function of excitation amplitude reveal the rich dynamics discussed by the authors.

I cannot suggest many changes that would improve the manuscript. Figure 3 could be improved if the distinction between linear LR dispersion curves vs nonlinear dispersion curves could be made more clear. Part of the issue is the superimposing of harmonic balance results with red markers obscures the blue line since many points are used. In fact, it looks like harmonic balance is following the linear results (solid blue line) better than the perturbation result (dashed blue line). I suspect the lines are mislabeled such that the perturbation result is solid blue, and the linear result is dashed blue.

1. Declaration of originality

We certify that this manuscript is original and not published or submitted for publication elsewhere. Actually, we report in our manuscript NCOMMS-17-06780 on several experimental and theoretical results on the transmission of elastic waves through one-dimensional (1D) and two-dimensional (2D) nonlinear acoustic metamaterials (*i.e.*, beam and plate). The experimental approach draws on the theoretical investigations of the band structures of **discrete** of nonlinear acoustic metamaterial models [47-49]. But we emphasize that the contents in NCOMMS-17-06780 and references [47-49] are absolutely different. More generally, our scientific approach is from ‘fundamental model’ to ‘complex model’: the present work falls within this context.

In our manuscript NCOMMS-17-06780 we demonstrate through theoretical and experimental studies that a nonlinear mechanism enables for greatly reducing the transmission of waves in an **ultra-low and ultra-broad** (double-ultra) band that we refer to as the “chaotic band”. Many interesting phenomena and properties are described in this manuscript, including the double-ultra wave suppression, the energy dispersion, nonlinear LR bandgaps, multi-states in the bandgap etc...How to explain these phenomena in such a complex nonlinear system?

Therefore, prior to this experimental work on the complex structures, we needed theories based on fundamental models to support the concept of chaotic band and to figure out the band structures in fundamental models, including bandgaps, passbands, bifurcations and chaos. Therefore, the discrete models published in references [47-49] lay some (although not all) theoretical foundations for the concept evidenced in NCOMMS-17-06780. All in all, the main results and implications reported therein are fully original and cannot be found in any of our previous works.

To the best of our view, the manuscript NCOMMS-17-06780 is the first work to report the double-ultra properties and demonstrate the novel mechanism ‘chaotic band’.

2. The mechanism and its application

Dear Referees, our understanding on the „mechanism and how it can be applied“ you care about contains three meanings: (1) What is chaotic band? (2) How do we realize the double-ultra effect with chaotic band? And (3) how can we use the chaotic band in practice? We try to explain them with the revised Fig.1, as shown below.

(1) What is chaotic band?

„Chaos“ is a concept that is relative to the concept „period“. Before the understanding of the chaotic band, we have to understand the band that consists of the periodic waves. For a periodic signal such as $\sin(2\pi f t)$, where f is the frequency and t is the time, the spectrum has only one peak (or pulse). In the physical view, this means all the energy is localized at the frequency f . A periodic input in the linear system generates a periodic output, as illustrated in Fig.1a. A resonance means all the energy is localized at the eigen-frequency and the corresponding eigen-mode. A linear dynamic system will be resonant at nature frequencies that would destroy the system. In the finite linear acoustic metamaterials, the broad multiple passbands actually consist of dense resonances, *i.e.*, resonant passbands, as you see in the linearized responses of the metamaterial beam and plate shown in Fig.3 and Fig.5. Though the bandgaps can suppress the wave propagation, they are narrow especially for the local resonant (LR) bandgaps. The generalized widths γ of the LR bandgaps we obtained are only $\gamma=0.1\sim 0.15$ (*i.e.*, $\gamma \ll 1$). Moreover, in the view of application, we are depressed by those dense resonances in the passbands that make the system become dangerous. That's why we start to study the nonlinear acoustic metamaterials.

In contrast, chaos is an aperiodic long-term behavior in a deterministic/nonlinear system exhibiting a strong dependence on the initial conditions. A chaotic signal is similar with a „random“ signal (see Fig.1a) whose spectrum would be continuous or at least contains multiple frequency components, as shown in Fig.8a. (In fact, chaos is not random; we call it „intrinsic randomness“.) Therefore the energy is not localized; instead, it is redistributed or dispersed to other broad bands. This effect becomes significant in the strongly nonlinear case that can „transfer“ the energy to other

frequencies and other parts to suppress the total amplitude.

Can we use the chaotic mechanisms to suppress the broadband resonances in the passbands of the acoustic metamaterials? To theoretically answer this question, we studied the discrete models in references [47-49], as stated in the first section. The band structure of a diatomic model is sketched in Fig.1a. In the discrete models, we found that the passbands of the nonlinear acoustic metamaterial (NAMs) models become a chaotic band, where a periodic input wave in this band generate chaotic output wave whose amplitude is much lower than the corresponding linear models.

Therefore, we give the concept of chaotic band: In nonlinear acoustic metamaterial, chaotic bands are the passbands where an incident low-frequency periodic wave (a slow wave) becomes a chaotic emerging wave that reduces the wave transmission. The chaotic wave features a high-frequency continuous spectrum (fast waves) evidencing the dispersion of energy. These waves have lower amplitudes than the corresponding linear resonances thus NAM can suppress the wave propagation in the passbands.

In the revised manuscript, we attempt to explain the concept of ‘chaotic band’ clearly with succinct sentences because of the length limitation. We believe readers will have a much deeper understanding of the ‘chaotic band’ when they read Fig.1, Fig.3a, Fig.5 and Fig.8 about the concept and experimental results.

Figure 1| Schematic and conceptual diagram. (a) Schematic of a diatomic NAM model composed of periodic nonlinear Duffing oscillators and linear base (left) and its band structure (right). (b) Conceptual diagram of the ultra-low- and ultra-broad-band mechanism using the chaotic band. The NAM cell contains two nonlinear sources: a Duffing oscillator and a coupled vibro-impact system (m_c couples

to J_r through a linear spring k_3 and a clearance δ). Two LR bandgaps is induced by them in the linearized model. There are four passbands, and the red and dashed black arrows represent the driving wave and its transmissibility in linear passbands, respectively; the thick green lines represent the upper limits of the chaotic bands, and the green arrows represent the frequency components in the chaotic response.

(2) How do we realize the double-ultra effect with chaotic bands?

The conceptual diagram of the band structures of our NAMs are shown in Fig.1b. The chaotic band has a property: the wave suppression effect of strong chaos is **broadband**, and it depends on the **location** but **not on the width** of the nonlinear LR bandgap. This property actually liberates us from the ‘width of bandgap’, which means we can suppress the broadband waves with a narrow bandgap. (Here, the ‘broad’ and ‘narrow’ are not contradict. They are used for different things.) Furthermore, it decouples the ‘location’ and ‘width’, which means we can realize a low-frequency effect. But in the linear regime, width and the location are coupled by the law $\gamma = \sqrt{1 + m_r/m_b} - 1$ (see Introduction).

Therefore, we can design a NAM with the nonlinear meta-cells that generate ultra-low-frequency but narrow linearized LR bandgaps. When strong nonlinearity occurs, the passbands higher than these nonlinear LR bandgaps become chaotic and wave propagation is suppressed—double ultra NAM is thus obtained.

To apply the ‘chaotic band’ mechanism for the double-ultra wave suppression in NAM, we need strongly nonlinear sources. As proof of concept, we designed both 1D and 2D NAMs to demonstrate the double-ultra properties. In the NAMs, a nonlinear LR bandgap induced by the relatively weakly nonlinear Duffing oscillator is designed in the ultra-low frequency; a coupled vibro-impact system is designed to generate strong nonlinearity and introduce the other low-frequency nonlinear LR bandgap. The two nonlinear sources expand the wave suppression effect in a broad chaotic band.

The first point in the third section details the techniques to design a strongly nonlinear cell. Moreover, we present rigorously theoretical and experimental proofs in the revised manuscript to demonstrate that the double-ultra wave

suppression effects are indeed induced by the chaotic mechanism (see the third section).

(3) How can we use the chaotic band in practice?

As explained above, resonances would destroy the structure and the chaotic band can suppress the wave propagation. The broadband resonances are common in civil structures such as cars, trains and airplanes. We built the NAMs based on a beam and a plate. As we know, beams and plates are widely used as the subject rods or shells in these civil structures. So actually, it is easy to use the NAM beam and NAM plate in these structures.

The application of the chaotic band is an open and interesting problem. We envision that the chaotic band induced double-ultra wave suppression open up new possibilities to the vibration and noise control, acoustic energy transfer and dissipation and elastic wave manipulation; and the multi-state behavior is benefit to control the state of acoustic devices.

We believe that, more, novel and maybe optimized nonlinear acoustic metamaterials will be built when the audiences and engineers know and understand the „chaotic band“.

3. Main changes to the manuscript

(1) Techniques to design the strongly nonlinear meta-cell

To apply the „chaotic band“ mechanism for the ultra-low and ultra-broad-band wave suppression in nonlinear acoustic metamaterials (NAMs), we need strongly nonlinear sources. At present, it is difficult but interesting to develop advanced techniques to reach the strong nonlinearities in macroscopic scale (Huang, P. *et al. Nat. Commun.* 7:11517 (2016)). In our NAM beam and NAM plate, we designed two nonlinear sources in the meta-cell: (i) the nonlinear stiffness coming from the magnetostatic field, *i.e.*, the Duffing oscillator; (ii) the collision that can be model as a vibro-impact absorber.

In the former manuscript, we made a statement on page 7 that “*In fact, there is a small clearance δ between the magnets and the strut. When the torsional amplitude is not small, the collision will occur in this clearance, which provide the other strongly nonlinearity sources in our NAMs. The theoretical method neglects this source.....*” This neglect helped us focusing on the double-ultra properties and the demonstration of the chaotic mechanism in the experiment.

Based on our studies, and to answer the questions of reviewers, to provide a rigorously theoretical demonstrations of the double-ultra effect and the chaotic mechanism, we take into account the designed collision in the clearance ($\delta=5 \times 10^{-4}$ m) in the theoretical methods of the revised manuscript. The collision is a vibro-impact oscillator, whose nonlinear interaction force is subject to the power law $P(x) = k_3x + k_c x^n$, $k_c = \alpha \delta^{-n}$ where $n > 1$ is an odd number and α depends on the estimated peak force (or acceleration) in the collision. The value of δ^{-n} is so large that the influence of α becomes weak, here $\alpha \approx 1$. k_c becomes very large as n increases, which can model the strong nonlinearity; for example, $k_c \approx 1 \times 10^{10}$ for $n=3$. Therefore the equivalent ‘nonlinearity stiffness’ of this vibro-impact system is much higher than the Duffing oscillator, which indicates that it can produce a strong nonlinearity under smaller amplitude than the former Duffing oscillator. (see *Methods* for more details.) The complete equivalent system is shown in Fig.2b (the figure below). Actually, the mass m_r generates two different nonlinear interactions in the motions along transverse and longitudinal directions, respectively. The two nonlinear effects act on different frequency ranges to reach the ultra-low and ultra-broad-band (double-ultra) effect, as shown in Fig.1b

In the revised manuscript, we compared the dispersion solutions and frequency responses of the NAM beam excluding

or including the vibro-impact oscillator, *i.e.*, system-N1 and system-N2, as illustrated in Fig.3b,c. The theoretical results agree with the experimental results better when we considered this strongly nonlinear motion. The influence of the exponent n is explained in Supplementary Note 3.4.

Because we have considered the influences of the influences of the vibro-impact oscillator, the conceptual diagram to design the nonlinear metamaterial was also revised. In theory, the nonlinear effect of k_2x^3 acts mainly on the frequency range 20-80 Hz, the nonlinear effect of $k_c x^n$ acts mainly on the frequency range higher than 80 Hz. The strength of nonlinearity depends both on the nonlinear stiffness coefficient (k_2 or k_c) and the amplitude. Though k_2 is much smaller than k_c when the exponent $n=3$, k_2x^3 can also generate significant nonlinear effect under moderate driving because the acting frequency of k_2x^3 is lower therefore larger average amplitude is obtained.

(2) Exclusion of other nonlinear sources --confirmation of the nonlinear effect

To confirm the double-ultra effect is induced by the Duffing and vibro-impact oscillators indeed, we have to exclude other nonlinear factors such as geometrical and inertial nonlinearities. To do so, we use both theoretical and experimental methods. (i) We establish the **nonlinear** finite element model of the 1D NAM beam that considers the geometrical and inertial nonlinearities in the primary beam, which proves that geometrical and inertial nonlinearities can be neglect if we had considered the nonlinear effect coming from Duffing and vibro-impact oscillators. (ii) We make another comparative experiment on a **linear** acoustic metamaterial beam, where the same levels of excitations (small and large excitations) are used to drive the linear acoustic metamaterial beam. As shown in supplementary Fig.S7 (the figure below), this comparative experiment demonstrate the double-ultra wave suppression in the NAM beam is induced by the nonlinear cells we designed. A comparative analysis on the responses of the NAM beam is also presented (Supplementary Fig.S8).

These theoretical and experimental results are detailed in the Supplementary Note 3.

Figure S7| Transfer functions of the LAM beam under small and moderate deformations. L1: the excitation voltage of the amplifier is 0.3V; L2: the voltage is 5.0V.

(3) Directly demonstration of the chaotic mechanism

All the work in this manuscript is to demonstrate that: the double-ultra properties come from the chaotic wave propagation in the NAM beam and plate.

In the former manuscript, we demonstrate the chaotic mechanism in experiment rigorously with Lyapunov Exponents (see Fig.8). However, the theoretical demonstration is indirect from the discrete models. To make the theoretical demonstration be more rigorous, we should present more directly proofs, *i.e.*, bifurcation of periodic solutions. However, it is not an easy work for the high-dimensional nonlinear systems (NAM beam and plate are high-dimensional systems). Fortunately, we can analyze the bifurcations of the periodic solutions of the NAM beam through the dimension-reduction algorithm combined with harmonic average approaches. The algorithm and its validity are described in Supplementary Note 4. By considering the vibro-impact oscillator, the bifurcations of the periodic solutions are illustrated in Fig. 3d,e (as shown below). The results indicate that the periodic solutions become unstable near the resonances, which would induce chaotic responses; and amplitudes of unstable solutions under increasing force would remain constant in the

chaotic band, which make the transmission decrease with the force.

(c) the theoretical response displacements at point A of the NAM and the corresponding LAM based on the FEM and HBM; in (c,e), the driving force at point E is $F = 5 \text{ N}$; (d,e) bifurcation diagrams under the force (d) and frequency (e). Thin black lines: solutions of LAM; solid blue: stable periodic solutions of the NAM; dashed green: unstable solutions.

For the NAM plate, we do not repeat the theoretical procedure. The regularities of bifurcations between the NAM beam and NAM plate are like.

(4) Multi-state of the NAM plate

To illustrate the multi-states in the bandgap of NAM plate, we present a contour plot of the distribution of the transmission difference in the whole plate, as illustrated in Fig.7b (the figure below). This result comes from the scanning experiment on the whole plate.

(5) Other main changes

To limit the length of the main text in 5,000 words in total, we

(a) move some contents in *Equivalent motions* in *Results* to the section *Methods*, and then we merge the sections *NAM design* and *Equivalent motions*.

(b) do our best to make all the sentences be clear and succinct.

4. Responses to referees

All the responses are highlighted with blue color.

Reviewer #1

The paper studies the nonlinear acoustic metamaterials with Ultra-low- and ultra-broad-band. The mechanism has not been explained clearly. Please describe the ultra-low and ultra-broad band results and give the physical reason.

Dear professor, thanks for your excellent advice. Following your comments, we realize that we should explain the mechanism clearly. In the revised manuscript, we hope we have explained the double-ultra effect, the chaotic mechanism and the methods to design our metamaterials clearly, as expounded in the second section in this letter and the part 'NAM design' in 'Results' in the main text.

Moreover, to give the physical reason of the double-ultra effect, we calculate the periodic solutions (i.e., the frequency responses) and the bifurcation properties of the NAM beam, as explained in point 2 and point 3 in the third section.

Because the length limit of the manuscript, we didn't repeat the theoretical procedure for the NAM plate.

Here are some questions in details:

1. What is the meaning of γ , is it generalized width or width-position ratio?

It is the generalized width that defined by the width-position ratio $\gamma=(f_{cut}-f_{st})/f_{st}$. We remove the ambiguous statement about the concept 'width-position ratio'. γ is the generalized width.

2. What is the meaning $a \approx 2\lambda/5$?

'a' is the lattice constant and λ refers to the wavelength at f_{st} .

3. What is the meaning of m_0 in Figure 1?

It should be m_b defined in the equation $\gamma = \sqrt{1 + m_r/m_b} - 1$.

4. Conceptual diagram of the ultra-low- and ultra-broad-band mechanism using the chaotic band. Question: How to realize the conceptual diagram using nonlinear acoustic metamaterials?

The method to realize the conceptual diagram is explained in the first point in the third section.

5.-chaotic band-reduce the wave transmission by as much as 20-40 dB. Question: The receiving signal is very small, it may be influenced by noise?

Actually, there are unavoidable noises in the signal, but we have taken both theoretical and experimental methods to confirm that the transmission reduction is induced by the ‘chaotic waves’.

(1) As detailed in the third section, we established a comparative experiment on the linear acoustic metamaterial beam and measured the responses in 0-2000 Hz under small and large excitations, respectively. The results show that the responses in the two situations are approximately equal, which demonstrates the signal-noise ratio is high enough.

Figure S8| Response velocities of the NAM beam under two excitations. L: case-i 0.1V in the main text; N: case-iv 5.8V in the main text.

Different cases are indicated in the legends.

(2) We directly compare the responses but not the transmissions of the nonlinear acoustic metamaterial, as illustrated in Supplementary Fig.S8 (the figure below). This analysis further proved the conclusion that the wave transmission is reduced by the nonlinear effects.

(3) In fact, the Largest Lyapunov Exponents (LLEs) shown in Fig.8 have also demonstrated the signal-noise ratio is high enough in our experiments. In theory, LLE approaches to $+\infty$ for a white noise signal. In the steady responses, we

measure the exciting signals as references for the responses signals. As shown in Fig.8, LLEs of the exciting signals are fluctuating near 0 and the maximum value is very small, which also demonstrates the signals are not polluted by the noises.

6. By contrast, the essence of the ultra-low- and ultra-broad-band (double ultra) NAM is enabled by the chaotic responses coming from the unique strongly nonlinear meta-cells.

Question: The chaotic response is hard to control?

Dear referee, this is an interesting and open question. I think we can understand it in this way.

Firstly, because the chaotic response is generated by the nonlinearity in the system, it is more difficult to analyze the chaos in theory than the linear system, especially for the high-dimensional system like our nonlinear acoustic metamaterials. To control the chaos, we need the bifurcation diagrams of the periodic solutions (see Fig. 3d,e in the revised manuscript).

Secondly, the challenge in the experiment is how to realize the strongly nonlinear sources, and we used two techniques to design our nonlinear metamaterial acoustic, i.e., the Duffing oscillator and the vibro-impact oscillator, as stated in the third section in this letter.

Fortunately, the nonlinear chaotic effect is broadband but not narrow-band as the resonances in linear systems, therefore it may be easier to control a chaotic response than a resonance in this view.

7. The nonlinear effect is broadband and it is independent of the width of the bandgap in NAM but it depends on the location of the nonlinear LR bandgap.

Question: Why the nonlinear effect is broadband and it is independent of the width of the bandgap in NAM? And the nonlinear effect is small.

Dear reviewer, we revise this statement as: The wave suppression effect of strong chaos is broadband, and it depends on the location but not on the width of the nonlinear LR bandgap.

We attempt to answer your question in several views.

(1) The conclusion ‘the nonlinear effect is broadband’ has been demonstrated in many books such as ‘Nayfeh, A. H. & Mook, D. T. *Nonlinear Oscillations*. (Wiley, New York, 1979)’. For a nonlinear vibration system, the frequency depends on the amplitude; therefore the ‘nature frequency’ can be modulated by the responses of the system itself, which make the nonlinear effect is broader than the linear resonance. Moreover, the essentially nonlinear oscillator can couple with any resonances in the system (Vakakis, A.F., et al, *Nonlinear Targeted Energy Transfer in Mechanical and Structural Systems*. Springer, Dordrecht (2008)), therefore the strongly nonlinear effect is broadband.

(2) The conclusion ‘The wave suppression effect of strong chaos is broadband’ was demonstrated in our work Ref. [49]. The effect of the chaos is coming from the effect of nonlinearity. In that work, we studied the varying laws of the Lyapunov Exponents in the discrete model, and the results indicate the strong chaos get a broadband wave suppression properties at least in the NAM models. This ‘broad’ is relative to the wave suppression effect of the ‘bandgap’. Results in this manuscript also demonstrate this property, as shown in Fig.3 to Fig.6.

(3) The conclusion ‘in NAMs, it is independent of the width of the bandgap’ was demonstrated in our work in references [47-49]. We didn’t find any direct proof that shows the wave suppression effect of chaos depends on the width of the bandgap. That is why we have to theoretically study the discrete fundamental models first. In the discrete model, the equations are numerically solvable and the responses can be simulated by numerical integrals, therefore we can understand those complicate phenomena.

(4) Dear referee, you are right that the nonlinear strength is weak in the former manuscript. That was because we neglected the vibro-impact motions in the metamaterials. By following your advices, we take into account the influences of this strongly nonlinear source in the revised manuscript. This revision is also explained in the third section.

8. What is the difference between the work and the references 47-49?

Following your advice, the differences between this work and the refernces [47-49] are stated in the first section 'declaration of originality'.

9. The experiments are proof-of-concept experiments. How to use it in practice?

The methods to use this mechanism in practice are discussed in the third point in the second section.

Reviewer #2

The paper presents new and interesting information. It can be published after introducing multiple corrections. I may suggest the following corrections:

1. Using unexplained symbols in the Abstract should be discouraged.

We remove the unexplained symbol λ in the abstract in the revised manuscript. Other symbols are defined. γ : the generalized width (the ratio of the bandgap width to its start frequency); 1D: one-dimensional; 2D: two-dimensional.

2. Some statements are not accurate and in fact they are misleading. For example, in page 1 authors write "In the nonlinear electromagnetic and photonic metamaterials [23,24], desired nonlinear responses [25,26] and many unique properties have been found and demonstrated [27-31]. However, nonlinear acoustic metamaterial (NAM) is a new topic, and their physical properties are waiting to be unveiled". But the situation is quite opposite. As a rule nonlinearity in electromagnetic and photonic metamaterials is weak unlike strong nonlinearity (and even nonlinearizable nonlinearity)

was observed in acoustic metamaterials, e.g. granular crystals described in [37, Chapter 1] and in many subsequent publications.

Thanks for your profound knowledge. The part introducing the nonlinear acoustic metamaterial is revised.

“Similarly to the nonlinear electromagnetic metamaterials^{23,24} where desired nonlinear responses^{25,26} have been demonstrated²⁷⁻³¹, nonlinear acoustic metamaterials (NAMs) deserve special attention. When acoustic waves propagate within a nonlinear acoustic media, such as Fermi-Pasta-Ulam (FPU) chains³²⁻³⁴ and granular crystals³⁵⁻³⁸, nonlinear phenomena including discrete breathers³⁹, solitons^{40,41} and bifurcations⁴² can be found. Acoustic diode^{36,43,44}, rectification⁴⁵ and lenses⁴⁶ have been designed based on the nonlinear media . However, the involved mechanisms hardly allow for simultaneous low-frequency and broadband properties and therefore, the discovery and development of new mechanisms are needed for further progress. ”

3. Some references incorrectly represent the state of the art. For example, in page 1, the statement “The presence of nonlinear media in linear periodic structure can be used to realize ultrasound acoustic diodes [32,33]” ignores the fact that acoustic diode was presented in the paper V.F. Nesterenko, et al., Physical Review Letters, vol. 95, pp. 158702, 2005 which was published four and five years before papers [32,33]. This fact is clearly confirmed by the citation from this PRL2005 paper “In summary we observed a strong sensitivity to the initial precompression of the reflected and transmitted energy from the interface of the two granular media. This phenomenon can be named the acoustic diode effect. It can be employed for designing tunable information transportation lines with the unique possibility to manipulate the signals delay and reflection at will, and decompositions/ scrambling of security-related information”.

Dear Referee, we have read the PRL2005 paper but here we missed its citation, because we were talking about the nonlinear media in linear periodic structure. The PRL2005 paper firstly found the diode effect for solitary wave in the strongly nonlinear media. In the revised manuscript, we refer this paper as [36].

4. In the text authors use ‘m_b’ but in the Fig. 1 they use ‘m₀’ which was not introduced in the text.

Yes, it should be m_b in Fig.1.

5. In the page 6 authors should clearly explain that expansion related to Eq. 1 is valid when amplitude of displacement is much smaller than the clearance (and not just smaller as in the text).

Dear Professor, from your comment, we made a complement about the theoretical description of the nonlinear acoustic metamaterials. Eq.1 is not enough to explain the strongly nonlinearity we have designed. The revisions are explained in point 1 and point 2 in the third section in this letter. Here we outline them again.

In the former manuscript, we made a statement on page 7 that *“In fact, there is a small clearance δ between the magnets and the strut. When the torsional amplitude is not small, the collision will occur in this clearance, which provide the other strongly nonlinearity sources in our NAMs. The theoretical method neglects this source.....”*

From your comment, we found that this simple treatment to neglect it would induce errors between the experiment and the theory. In the theoretical methods of the revised manuscript, we take into account the collision in the clearance ($\delta=5 \times 10^{-4}$ m). The collision is a vibro-impact oscillator, whose nonlinear interaction force is subject to the power law $P(x) = k_3x + k_c x^n$, $k_c = \alpha \delta^{-n}$ where $n > 1$ is an odd number and α depends on the estimated peak force (or acceleration) in the collision. The value of δ^{-n} is so large that the influence of α is weak, here $\alpha \approx 1$. k_c becomes very large as n increases, which can model the strong nonlinearity; for example, $k_c \approx 1 \times 10^{10}$ for $n=3$. Therefore it can produce a strong nonlinearity under a smaller amplitude than the former Duffing oscillator. (see the section ‘*Equivalent motions*’ in **Methods** for more details.) The complete equivalent system is shown in Fig.2b (the figure below). Actually, the mass m_r generates two different nonlinear interactions in the motions along transverse and longitudinal directions, respectively. The two nonlinear effects act on different frequency ranges to reach the ultra-low and ultra-broad-band (double-ultra)

effect.

In the revised manuscript, we compared the dispersion solutions and frequency responses of the NAM beam excluding or including the vibro-impact oscillator, *i.e.*, system-N1 and system-N2, as illustrated in Fig.3b,c. The theoretical results agree with the experimental results better when we considered this strongly nonlinear motion. The influence of the exponent n is explained in Supplementary Note 3.4.

Because we have considered the influences of the influences of vibro-impact oscillator, the conceptual diagram to design the nonlinear metamaterial was also revised. In theory, the nonlinear effect of k_2x^3 acts mainly on the frequency range 20-80 Hz, the nonlinear effect of $k_c x^n$ acts mainly on the frequency range higher than 80 Hz. Though k_2 is much smaller than k_c when the exponent $n=3$, k_2x^3 can also generate significant nonlinear effect under moderate driving because the acting frequency of k_2x^3 is lower therefore larger average amplitude will obtain.

Other nonlinear factors such as the geometrical nonlinearity and inertial nonlinearity are proved neglectable in the amplitude range we studied. This demonstration is expounded in Supplementary Note 3.

Moreover, we made another comparative experiment on the linear acoustic metamaterial beam to demonstrate the nonlinear effect (see Supplementary Fig.S6). Directly demonstration of the chaotic band is explained in point 3 in the third section.

6. The sentence in the page 9 “Further increasing the driving amplitude strengthens the nonlinearity to be non-negligible”

is awkward. Probable replacement would be “Further increasing the driving amplitude strengthens the nonlinearity which can’t be neglected”.

We remove this sentence because we have defined the strength of nonlinearity to illustrate the nonlinearity clearer, as stated in point 8 below.

7. The statement in page 10 “The responses of the NAM are amplitude-independent in practice” is confusing and makes the subject of the whole paper questionable. Indeed the focus of the paper is on the nonlinear acoustic metamaterials and nonlinearity is supposed to make responses of NAM amplitude dependent!

We remove the statement “The responses of the NAM are amplitude-independent in practice” in page 10. And in page 8, we state that “The coupled periodic nonlinear oscillators create amplitude-dependent properties in NAMs.”

8. Authors use terms “strongly nonlinear system” (page 10), “stronger nonlinearity”, “different nonlinear strengths” (page 12), “strong nonlinearity” (page 14), “unique strongly nonlinear met-cells”, “strong nonlinearity” (page 19). It is not acceptable unless authors attribute some numbers to this terms. The possible suggestion would be the characteristic values of the ratio of two terms (linear and nonlinear) in Eq. 1 for corresponding cases.

As detailed in question 5, the nonlinearity of the vibro-impact oscillator is stronger. Therefore the nonlinearity of the whole system can be estimated based on this system. In the revised manuscript, we define the strength of nonlinearity σ through the stiffness ratio between nonlinear part and linear part. From Eq.2, The strength of nonlinearity of the NAMs is defined by $\sigma = nk_c A^{n-1} / k_3$, where A stands for the response amplitude [47,49]. σ is a relative indicator in our experiments. When $\sigma \approx 0$, it is the approximately linear case. For the experiments on the NAM beam, there are {i: $\sigma = 7.5 \times 10^{-4} \approx 0$ }; {ii: $\sigma = 0.028$ }; {iii: $\sigma = 0.9262$ }; and {iv: $\sigma = 4.22$ }, i.e., the nonlinearity strength σ increases from 0 (linear) to 4.22 (strongly nonlinear). For the experiments on the NAM plate, there are {i: $\sigma = 0.0067 \approx 0$ }, {ii: $\sigma = 0.157$ }, {iii:

$\sigma=0.669$ and $\{\text{iv}:\sigma=3.21\}$, which also indicates that the nonlinearity increases from weak to strong. These analyses prove the definition of σ is reasonable.

This definition is provided in the section *Methods*

9. Finally, the English in the paper should be polished.

The English in the revised manuscript is polished to be native and scientific. Thank you very much!

Reviewer #3

This paper performs novel analytical and experimental analyses to uncover chaotic bands in one-dimensional (beams with attached nonlinear resonators) and two-dimensional (plates with similar attachments) systems. The systems considered, incorporating magnetically-levitated masses, are a clever means for achieving the desired nonlinearity. The analysis of this system is rigorous and thorough, and convincingly captures the system dynamics. Subsequent studies of the wave propagation as a function of excitation amplitude reveal the rich dynamics discussed by the authors.

Dear Professor, thanks very much for your comment. We have to mention that, in our meta-cell, the mass m_r is designed to generate two different nonlinear interactions in the motions along transverse and longitudinal directions, respectively. In the former manuscript, we neglected the vibro-impact oscillator in the theoretical method. In the revised paper, we take into account it and provide more direct demonstrations for the new mechanism. These revisions will make the physics of this paper be more rigorous.

I cannot suggest many changes that would improve the manuscript. Figure 3 could be improved if the distinction between linear LR dispersion curves vs nonlinear dispersion curves could be made clearer. Part of the issue is the superimposing of harmonic balance results with red markers obscures the blue line since many points are used. In fact, it

looks like harmonic balance is following the linear results (solid blue line) better than the perturbation result (dashed blue line). I suspect the lines are mislabeled such that the perturbation result is solid blue, and the linear result is dashed blue.

Yes, the dispersion plot in Fig.3b is not clear in the former manuscript. We revised it as below. Moreover, the perturbation approach is proper for weakly nonlinear system. For the strongly nonlinear system, it would present wrong results, on which occasion harmonic balance method should be adopted. Here, we present the perturbation results just for references, and the harmonic balance results are more accurate.

(b) Half Dispersion curves of the system-N1 (left) and N2 (right). κ : the wave vector, $0 \leq \kappa a \leq 2\pi$, and solutions are symmetrical respect to π for the 1D NAM. In (b), blue lines: LAM; red lines: the NAM solved with the harmonic balance method (HBM) with displacement amplitude in case iv; black dashed line: the corresponding perturbation results. In (b- Left), only the first dispersion curve of the perturbation results and three curves of HBM result are shown because the others superpose with the linear results. The three different shading colors denote the linear bandgaps. LR1 (LR2) represents the first (the second) LR bandgap.

Moreover, in Fig.5b, the dispersion solutions for the case N2 are not presented because: (1) as the results of NAM beam show, the two cases are similar; (2) perturbation approach is unsuitable for the strongly nonlinear system N2 but HBM is difficult to obtain all the analytical solutions of the high-dimensional system of equations.

REVIEWERS' COMMENTS:

Reviewer #1 (Remarks to the Author):

The authors have answered some questions I raised. However, please pay attention to the following points:

1. The main point is that the main theoretical results have been published in the references [47-49].
2. Chaos is an aperiodic long-term behavior in a deterministic/nonlinear system exhibiting a strong dependence on the initial conditions. Therefore it is hard to use it in practices.
- 3 Chaotic band is hard to explain the broadband of system.

Reviewer #2 (Remarks to the Author):

Authors made significant changes in the original text. The paper can be published.

Reviewer #3 (Remarks to the Author):

The reviewer's comments have been adequately addressed.

Responses to Reviewers

Dear Referees,

Thanks very much for wise comments and advices in all your reviews. In this letter, we answer your comments and questions point by point. Responses are highlighted with blue color.

Reviewer #1 (Remarks to the Author):

The authors have answered some questions I raised. However, please pay attention to the following points:

Dear Reviewer, thanks very much for your excellent advices and questions, which guide us to think this topic again.

1. The main point is that the main theoretical results have been published in the references [47-49].

Dear Reviewer, we certify that this manuscript is original and not published or submitted for publication elsewhere. The main results and implications reported therein are fully original and cannot be found in any of our previous works.

As you know, from 'fundamental model' to 'complex model' is the general way we understand physics. The present work also falls within this context. Nonlinear acoustic metamaterial is a new/young topic. We believe hundreds of papers about this topic are deserved to figure out the plentiful dynamics in future. Moreover, the 1D and 2D nonlinear acoustic metamaterials (beam/plate) we fabricated are **complex high-dimensional nonlinear** systems. It is difficult to explain the chaotic regimes in only one paper from fundamental theories to applications, rigorously and clearly.

Therefore, in references [47-49], we made theoretical studies on one-dimensional (1D) **discrete diatomic and tetratomic** models to support the concept of chaotic band and to figure out the band

structures in nonlinear acoustic metamaterial models. However, the contents in **NCOMMS-17-06780** and references [47-49] are absolutely different. The results in Refs. [47-49] tell us the concept of chaotic band and characteristics of chaos in discrete models, but they **do not clearly and directly** tell us that the chaotic band can achieve ultra-low and ultra-broad band wave reduction and suppression. Actually, we get this important property of chaotic band from the experiments on our metamaterial beam and plate. And then, we establish the theoretical model (the finite element model and the equivalent approach) of the nonlinear metamaterial beam/plate. The band structures are different from the discrete model. Both the periodic solutions solved from the theory and the largest Lyapunov exponents calculated from the wave signals in experiments demonstrate the double-ultra property derives from the chaotic band. **Therefore the demonstrations in our work NCOMMS-17-06780 can be independent from other publications.** However, the analyses in Refs. [47-49] validate some algorithms (such as the harmonic average approach and perturbation continuation method) and provide general understandings on the chaos in nonlinear acoustic metamaterials (such as why the largest Lyapunov exponent in the first passband is so small). These analyses help making the theories and experiments in the manuscript **NCOMMS-17-06780** be more rigorous, more credible and more understandable.

Another significance of the work **NCOMMS-17-06780** is that we design a kind of structure of nonlinear meta-cell and fabricate the nonlinear acoustic metamaterial beam and plate, which at last demonstrates the double-ultra property. This work tells us nonlinear acoustic metamaterials are realizable in practical structures such as beam and plate.

2. Chaos is an aperiodic long-term behavior in a deterministic/nonlinear system exhibiting a strong

dependence on the initial conditions. Therefore it is hard to use it in practices.

Dear reviewer, you are right. It is challenging to use chaos in practices because of the complex behaviors and the sensitivity to initial conditions. But the challenge will become a reality if we figure out three questions: what property of chaos can be used to realize what functions; how to design a strongly nonlinear structure to generate strongly chaos; what does the sensitivity mean in nonlinear acoustic metamaterials. Through the theoretical and experimental studies in this paper, we make it clear that the mechanism chaotic band is promising in ultra-low and ultra-broad-band wave control.

In this paper, we use three properties of chaotic waves propagating through nonlinear metamaterials: the chaotic wave can disperse energy in a broad high-frequency band but not localize energy to generate resonances; the amplitude of a chaotic wave is bounded; chaotic effect is broadband but not localized at a special frequency. The three properties tell us chaos can be used in broadband wave reduction and suppression. There is another useful property of chaotic band in nonlinear acoustic metamaterials: the chaotic effect depends on frequencies but not depends on (at least not strongly depends on) the width of the nonlinear bandgaps. This property liberates us from the width of the bandgap and so decouples the 'low-frequency' and 'broad' of a band for wave suppression. Therefore it can be used to realize a low-frequency but broad band. Moreover, we design a nonlinear meta-cell that can generate strong chaos in the practical structures: beam and plate. This meta-cell can be a reference in the field nonlinear acoustic metamaterials. And the beam and plate are practical structure in civil engineering.

Moreover, 'Chaos is an aperiodic long-term behavior in a deterministic/nonlinear system exhibiting a strong dependence on the initial conditions' is a general definition of chaos. Chaos is sensitive to initial conditions. This is another difficulty in practice. In this paper, the sensitivity presents as the

dependence on amplitudes. When we say ‘sensitivity’ of chaos, generally we means the long-term trajectory or the process evolving in time domain depends strongly on the initial conditions, thereby a long-term prediction is impossible. But not all the characteristics of chaos are sensitive to initial conditions. This is an open topic. In the view of ‘control’, it would be difficult sometimes if we want to control a specified parameter or state with chaos because of the sensitivity, but sometimes chaos helps to make the controlling become robust, for example, in the frequency range 300-700 Hz in Fig. 5, the wave suppression is robust to the driving amplitudes. Another example is shown in Ref. [49]: under some frequencies, the average amplitude of chaos would keep constant under the increasing driving.

As you mentioned, the topic “nonlinear acoustic metamaterial” is not as easy as linear metamaterials whose behaviors can be predict from the unique solution. Our work attempts to make the difficulty become understandability. This is what the scientists around the world are doing, right?

Our studies figure out the properties of chaotic band and the experimental results show that this mechanism is promising in in ultra-low and ultra-broad-band wave reduction and suppression. The fabricated structure and the experimental results also give us much confidence to make further studies on the dynamics of nonlinear acoustic metamaterials and to develop applications of them.

3. Chaotic band is hard to explain the broadband of system.

Dear reviewer, this is a very good question.

In practice, we have tried to explain the ‘broadband’ of the systems with other mechanisms, such as, the **nonlinear resonances**. For example, we make a statement in page 10: the resonances in the second and the third passbands are significantly reduced because the linear resonances are replaced by the **nonlinear resonances** with finite amplitudes.

When the nonlinearity is weak or in a narrow frequency range below LRI, periodic solutions are stable and the nonlinear resonances can explain why the transmissions are reduced. However, under strong nonlinearity, theoretical results show the solutions near the resonances become unstable, which means **chaotic responses**. The largest Lyapunov exponents coming from the experiments also demonstrate the responses **become chaotic**. Moreover, this phenomenon appears in a broadband, as illustrated in Fig. 3e,f and Fig. 8. Therefore, **the essence** of the nonlinear resonances in the broadband under strongly nonlinear conditions **is chaotic responses**.

Therefore, based on both the theoretical and experimental demonstrations, at present, we find that chaotic band is the best explanation of the broadband of the systems.

Reviewer #2 (Remarks to the Author):

Authors made significant changes in the original text. The paper can be published.

Dear reviewer, thanks very much again for your wise and profound advices provided last time.

Those advices help improving this manuscript so much!

Reviewer #3 (Remarks to the Author):

The reviewer's comments have been adequately addressed.

Dear reviewer, your encouraging comments and advices help us going further on the topic nonlinear acoustic metamaterials. Thank you very much!